# What can entropy metrics tell us about the characteristics of ocular fixation trajectories?

**Kateryna Melnyk** *, Lee Friedman, Oleg V. Komogortsev

Department of Computer Science, Texas State University, San Marcos, TX, United States of America

* k_m825@txstate.edu

## Abstract

In this study, we provide a detailed analysis of entropy measures calculated for fixation eye movement trajectories from the three different datasets. We employed six key metrics (Fuzzy, Increment, Sample, Gridded Distribution, Phase, and Spectral Entropies). We calculate these six metrics on three sets of fixations: (1) fixations from the GazeCom dataset, (2) fixations from what we refer to as the "Lund" dataset, and (3) fixations from our own research laboratory ("OK Lab" dataset). For each entropy measure, for each dataset, we closely examined the 36 fixations with the highest entropy and the 36 fixations with the lowest entropy. From this, it was clear that the nature of the information from our entropy metrics depended on which dataset was evaluated. These entropy metrics found various types of misclassified fixations in the GazeCom dataset. Two entropy metrics also detected fixation with substantial linear drift. For the Lund dataset, the only finding was that low spectral entropy was associated with what we call "bumpy" fixations. These are fixations with low-frequency oscillations. For the OK Lab dataset, three entropies found fixations with high-frequency noise which probably represent ocular microtremor. In this dataset, one entropy found fixations with linear drift. The between-dataset results are discussed in terms of the number of fixations in each dataset, the different eye movement stimuli employed, and the method of eye movement classification.

## Introduction

In 2018, our research group introduced an extensive set of eye movement features [1]. This set includes features such as fixation duration, saccade peak velocity, and the percentage of saccades that were followed by post-saccadic oscillations (PSOs). Since then, this work has been used in oculomotor biometric applications by our group [2–5] as well as by other research groups [6–8]. Furthermore, these features have been used in various other applications, including the development of biomarkers [9], gender discrimination [10], and aging [11].

In the current study, we introduce an entirely new set of features based on various entropy measures. We applied these features to fixation trajectories, and in general, by using them, we address the question: "Which fixation trajectories are complex in some way, and which are not?"

**Data Availability Statement:** All data are available at publicly available website: https://digital.library.txstate.edu/handle/10877/16142.

**Funding:** (1) O.V.K., National Science 608Foundation, CNS-1250718 and CNS-1714623,

www.NSF.gov; (2) O.V.K., National Institute of 609Standards and Technology, 60NANB15D325, www.NIST.gov; (3) O.V.K., National Institute of 610Standards and Technology, 60NANB16D293, www.NIST.gov; he funders had no role in study design, data collection and analysis, decision to publish, or preparation of the manuscript.

**Competing interests:** The authors have declared that no competing interests exist.

The chaotic nature of eye movement trajectories was first evaluated 30 years ago [12]. As far as we are aware, no one has addressed the question: "What qualities of ocular fixation can be detected using chaos-based metrics?"

Chaos theory [13–15] provides numerous metrics (see an overview Table 1) to assess different types of the complexity of an observed event. The chaos analysis developed rapidly and began to be applied in various biological [16] and social [17] science contexts. Chaos and entropy analysis have been applied to several forms of time series, including electrocardiograms (EKG) [18], electroencephalograms (EEG), and magnetoencephalography recordings (MEG) [19], as well as arterial blood pressure time series [20]. These methods were found to enhance the characterization [21] or diagnostic relevance of these signals [22, 23]. One interesting paper employed Independent Component Analysis (ICA) and Renyi Entropy to detect artifacts in EEG signals acquired during signal acquisition [24].

## Saccades

One interesting study [25] used chaos theory for the analysis of saccadic eye movements of healthy subjects. They assessed the complexity degree in the saccade dynamics by some computational tests (Power Spectrum, portrait in the state space, Fractal Dimension, Hurst Exponent, and Largest Lyapunov Exponent) and showed that there was a chaotic dynamic trend in saccade behavior. In their next study [26], they used metrics from chaos theory to evaluate saccades under two conditions: a saccade tracking task without distraction and one with distraction (cognitive load). The chaos of saccades was assessed with various metrics

**Table 1. An overview table presenting core chaos and entropy measures.**

| Feature | Meaning |
|---|---|
| Hurst Exponent | Quantifies the long-term memory of a time series, indicating how much it deviates from a random walk. Its values range from 0 to 1, with H = 0.5 representing a geometric random walk. |
| Largest Lyapunov Exponent | Evaluates time series predictability. Higher values indicate greater chaos and unpredictability, while lower values suggest increased stability and regularity in the time series. |
| Shannon, Fuzzy, Increment, Sample, Gridded Distribution, Phase, Spectral Entropy | They all measure the level of randomness or uncertainty in a time series, but they differ in their calculation methods and how they capture patterns in the data. The interpretation of the results depends on the data being analyzed. In general, higher values for entropy indicate more randomness, while lower values indicate a higher level of predictability or structure in the time series. |
| Independent Component Analysis (ICA) | The resulting components represent distinct sources of information that correspond to different underlying processes in the data. The interpretation of these components depends on the specific domain of the time series analysis |
| Fractal Dimension | Measures the roughness or smoothness of time series data. It can vary from a minimum value representing extreme smoothness to a maximum value indicating high roughness. The range of values depends on the data being analyzed. |
| Correlation Dimension | Measures the similarity or correlation between points in a time series. A higher value indicates nonlinear behavior, while a lower value suggests a more regular pattern in the time series. |
| Spectral Density Analysis (SDA) | Measures the power distribution across frequencies. The power distribution helps identify dominant frequencies, periodic patterns, or oscillatory behavior present in the data. The interpretation of the results depend on the objectives of the analysis. |

under both no-load and load conditions. It was found that the saccades during the no-load condition were more chaotic than saccades during the load condition.

Another study used a wavelet analysis of eye movements, collected during scene-viewing, to remove artifactual "spikes" that occur during signal acquisition [27]. Then they evaluated patients with cerebellar disorders as compared to healthy subjects. They found that Wavelet Entropy was a valuable measure of physiological motor noise.

Another group was interested in applying entropy measures to a visual scanning task to assess "visual scanning efficiency" [28]. The equation for Shannon Entropy was used to create the metric "Stationary Gaze Entropy" (SGE). The equation for Conditional Entropy was used to create "Gaze Transition Entropy" (GTE). They present a model of gaze orientation, with GTE as a measure of visual scanning efficiency and SGE as a measure of gaze dispersion.

This same group of authors [29, 30] looked at the usefulness of SGE and GTE to assess visual scanning while intoxicated. In the first study [29] they studied the effect of alcohol on gaze behavior and on measures of entropy. In the later paper [30] the authors evaluated whether alcohol's effect on gaze behavior can be detected during a simulated driving task. They found that as blood alcohol level was either increasing or decreasing there were statistically significant changes in SGE and GTE. The fixation rate was not affected by these conditions, however.

In [31], the authors measured the optimum phase space parameters ($\tau$, and $m$) for a series of eye movement velocity recordings collected while subjects tracked a jumping point target. The velocity records included the saccade to a new target and the fixation on that target. Recordings were obtained for unfiltered data and data filtered by one of 7 low-pass filters. Larger $\tau$ values and $m$ values were associated with more restrictive filters. The authors also found evidence of a long-range correlation in eye-tracking data. This can be interpreted as indicating that the signal is "self-similar" and can be considered a fractal signal.

Two papers focused on the usefulness of chaos and entropy-related features in eye movement classification [32, 33]. Harezlak et al. [32] reported that Approximate Entropy on various time scales might be useful for saccade detection. In a follow-up report [33] they found that Approximate Entropy, Fuzzy Entropy, and the Largest Lyapunov Exponent were more useful in eye movement classification. However, the percent correct classification in the best case was in the low 80%. This is probably not competitive with current state-of-the-art eye movement classification systems [34].

## Fixations

In the eye-tracking field, a number of reports have already evaluated chaos- and entropy-related metrics from fixational eye movements [35]. Their database consisted of fixations of three lengths, 1000, 1500, and 2999 ms duration. They investigated the correlation dimension (CD) of these 3 fixation durations before and after noise removal by a digital low-pass filter (50 Hz cutoff) of the eye movement signals. They found that the CD decreases with increasing fixation duration. They also found that the CD for the 1000 and the 1500 ms durations increased for low-pass filtered signals.

One paper [36] focused on fixations during the viewing of a complex scene. They did not analyze fixation trajectories. Rather, they examined the relationship between the dispersion parameter used in some fixation detection algorithms and the number of fixations detected. Obviously, as the dispersion threshold increases, fewer fixations are found. This relationship follows a power law. From this relationship, they present a method to measure the fractal dimensionality of a fixation. They note that the scanning patterns of typically developing toddlers may have fractal qualities.

Fixation eye movements from patients with schizophrenia were compared to those from a normal control group during reading [37]. The eye movement signals themselves were not analyzed. Rather, the focus was on the distribution of fixation durations. The authors found that the cumulative distribution of fixation durations from the patient group was a linear curve, suggesting that these fixations are a "temporal fractal". This was not found for the control group.

In [38], the researchers studied the relationship between images with various levels of complexity (higher "fractality") and the fixations recorded while viewing them. They found that there was a significant effect of the stimuli and that stimuli with higher complexity (higher fractality) cause fixational eye movements with lower fractality.

In [39], the authors evaluated both power spectral density analysis and fractal dimension determination from reading eye movements. They make two conclusions: (1) the underlying mechanism for eye movements might be effectively modeled by a chaotic, nonlinear, dynamical system possessing a strange attractor with a non-integer dimension. (2) The correlation dimension of the attractor, D2, an index of the complexity of the motion, seems to be related inversely to the qualitative assessment of reading ability.

## Scanpaths

Scanpaths are eye movements collected by a gaze-tracking device, where information is recorded about the trajectories (paths) of the eyes when scanning the visual field (areas of interest or AOIs) and viewing and analyzing any kind of visual information. Many studies have related scanpaths to cognitive abilities such as learning, discrimination, and recognition of AOIs [40–43]. Some of this research has indicated that scanpaths are essential to visual learning and recognition [44, 45]. Entropy-related metrics have been applied to evaluating scanpaths [46–51].

Entropy measures of scanpaths have been used in studies of autism, attention-deficit/hyperactivity disorder (ADHD), and anxiety. In autistic patients, entropy metrics have been used to evaluate scanpaths during face recognition [47]. The researchers evaluated patients with autism and healthy controls and found the entropy metrics suggested different developmental trajectories for the two groups. In another study, patients with ADHD were compared to a group of age-matched typical children [48]. Scanpaths were evaluated as subjects viewed words they were instructed to remember. The ADHD group did not follow typical scanpaths while viewing the words. Rather, their visual scanning was discontinuous, uncoordinated, and chaotic. In another study [49], subjects had to land an airplane in a simulator during normal conditions and under various types of anxiety. Visual scanning entropy, which is the predictability of visual scanning behavior, showed an increase in the randomness of scanning behavior when anxious.

The creators of marketing material often use particular visual cues or stimuli to bias the gaze toward a specific direction. This is referred to as gaze guidance. In the work of Hooge and Camps [50], a measure of scanpath entropy was able to assess the effectiveness of gaze guidance [50].

Krejtz et al. [51] evaluated the usefulness of normalized Shannon Entropy during gaze transitions between AOIs. This metric expresses the complexity of both transition and stationary distributions between AOIs. They showed that higher transition entropy denotes more randomness and more frequent switching between AOIs. Also, they showed that higher stationary entropy indexes the extent to which individual visual attention is distributed among AOIs. Thus, they conclude that entropy measures are promising eye movement indicators of individual differences in curiosity, interest, and AOI familiarity.

### In the present study

Most prior research has focused on only a small set of chaos and entropy metrics. No prior report has evaluated several entropy metrics simultaneously for fixation time series specifically and exclusively. In the present study, we evaluated six entropy metrics as applied to fixation trajectories. For fixation eye movement signals, we employed three datasets: (1) GazeCom [52, 53]; (2) the "Lund" dataset [54], and a dataset from our laboratory ("OK lab") [55]. In our Methods section we endeavor to impart a more intuitive understanding of what aspect of a fixation signal that each entropy metric is sensitive to. To evaluate the types of fixations that high and low values of our metrics found, for each entropy measure we evaluated a page of 36 fixations with the highest entropy values and a page with 36 fixations with the lowest entropy values. Our overall aim is to determine what entropy metrics can inform us about fixation trajectories. With three datasets, we can evaluate if this information is dataset-specific.

## Methods

### Description of the GazeCom dataset

The GazeCom data set consists of eye movement recordings at 250 Hz from an SR Research EyeLink II device.

From [53], who employed the GazeCom dataset:

The GazeCom dataset consists of 18 clips (videos), around 20s each, with an average of 47 observers per clip (total viewing time over 4.5h). The total number of individual labels is about 4.3 million (including the samples still recorded after a video has finished; 72.5, 10.5, 11, and 5.9% of all samples are labeled as parts of fixations, saccades, pursuits, or noise, respectively). Event-wise, the data set contains 38,629 fixations, 39,217 saccades, and 4,631 smooth pursuits.

In this dataset, the horizontal position is measured in degrees of visual angle from 0 to 50 degrees. Zero represents the extreme left of the screen, 50 represents the extreme right of the screen and 25 degrees represents the left-right center of the screen (Mikhail Startsev, personal communication).

A full description of the dataset, the purpose of the dataset, the types of stimuli, and the recording details are provided in [52] and also [53]. We chose GazeCom, in part, because of the large number of identified fixations.

We derive our information as to how saccades were classified in Gazecom by reference to several papers [52, 53, 56]. Here is how [52] describes the process:

To initialize the search for a saccade onset, velocity had to exceed a relatively high threshold (138 deg/s) first. Then, going back in time, the first sample was searched where velocity exceeded a lower threshold (17 deg/s) that is biologically more plausible but less robust to noise (both parameters were determined by comparing detection results with a hand-labeled subset of our data). In a similar fashion, saccade offset was the first sample at which velocity fell below the lower threshold again. Finally, several tests of biological plausibility were carried out to ensure that impulse noise was not identified as a saccade: minimal and maximal saccade durations (15 and 160 ms, respectively) and average and maximum velocities (17 and 1030 deg/s, respectively).

Here is some additional information from [56]:

The two novice annotators were paid undergraduate students who received basic instructions about eye movements and interpreting eye tracker data. Experts in the eye movement field were available to answer their questions at any point in the labeling process. Due to their little prior experience with hand-labeling and because we wanted their internal biases to stabilize, these two annotators went through the data set for a second time several months later. In the first pass, they were provided with the prelabelled suggestions from the automatic method just described and were instructed to change, add, or remove intervals accordingly. In the second pass, they were presented with their own labeling and instructed to change it wherever they thought it was not accurate (with respect to the eye movement definitions). As a quality assurance measure, a third (expert) annotator (one of the authors) re-examined all the recordings in the data set with the objective of resolving conflicts between the labels of the first two annotators, also making changes where the provided eye movement definitions were violated.

It is worth noting that the post-saccadic oscillations (PSOs) were not classified separately in the GazeCom dataset, and yet, PSOs do occur in this dataset. In such a case: (1) the PSO is considered part of the saccade, or (2) the PSO is considered part of the fixation, or (3) some part of the PSO is considered part of the prior saccade and some part of the PSO is considered part of the following fixation. We note when a fixation begins with PSO or a part of a PSO.

One fixation was lost in the interpolation step (described below). This left 38,628 fixations. The median duration for a fixation was 60 samples. We also rejected fixations that were longer than an outlier threshold determined by the generalized extreme Studentized deviate test [57]. On this basis, fixations longer than 286 samples were outliers (N = 711). Of the remaining fixations, 831 fixations were rejected because there were not 30 samples of data before the fixations and 876 were rejected because we did not have 30 samples of data after the fixations. For our entropy calculations, we needed more than 10 samples. A number of fixations (N = 95) were rejected because they were less than 11 samples in length. During calculation of the entropies, if any fixation produced any entropy value less than or equal to 0.0, NaN ("Not a number") or was infinite, the entire fixation was dropped from the analysis (N = 465). After these screening steps, there were 35,650 fixations available. We present a violin plot of all fixations lengths, in samples, in Fig 1(A).

Although the nominal sample rate of the GazeCom dataset is 250Hz (an inter-sample interval (ISI) of 4 ms), there are very many unique ISIs other than 4 ms. Of all ISIs (4,227,557) 93.9% had the correct ISI (4 ms). Prior to the analysis of a dataset, the time stamps, horizontal position values, and classification codes were interpolated to a 4 ms ISI.

## Description of the Lund dataset

What we refer to as the "Lund" dataset was first described in Larsen et al. [54]. It used to be publicly available but we don't know if it still is. Interested users may want to communicate with Marcus Nyström (email: marcus.nystrom@humlab.lu.se).

This dataset was employed by Zemblys et al [58]. Here is how these authors described the dataset:

1cm "It consists of monocular eye movement data of participants viewing images, videos, and moving dots. The eye movements of all participants were recorded with the SMI Hi-Speed 1250 eye-tracker, running at a sampling frequency of 500 Hz. Two domain experts then manually segmented data into fixations, saccades, post-saccadic oscillations (PSO), smooth pursuit, blinks, and undefined events. A comprehensive description of the Lund dataset and the coding process can be found in [54]." (page 3, left column, last paragraph).

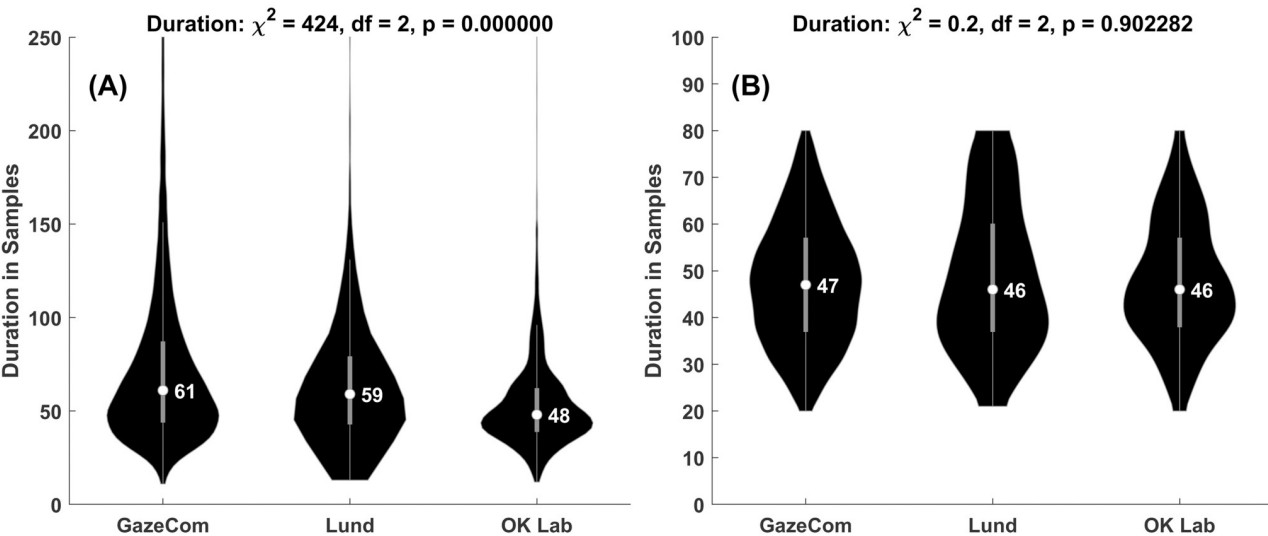

**Fig 1. Fixation durations across the three datasets.** In Fig 1 violin plots are presented representing the duration of fixations (in samples) across the three datasets. The durations between datasets were statistically significantly different. In Fig 1B we show the same plots after we had taken steps to equalize the fixation duration distributions across datasets.

The two domain experts that classified this data were Marcus Nystrom and Richard Andersson, both considered experts in the classification of eye movements. There were 34 recordings where subjects were observing static images for 10 seconds. Although the description of the dataset states that the dataset was collected with a sampling rate of 500 Hz, Friedman [59] reported that 6 of the recordings were actually collected at 200 Hz. After the removal of these recordings, 28 recordings were left. The largest number (16) of good datasets (i.e., collected at 500 Hz) were classified by Andersson. Since we were not interested in comparing raters, we only evaluated these 16 recordings.

These recordings were classified into 6 categories: (1) Fixation, (2) Saccade, (3) Post-saccadic oscillation, (4) Smooth pursuit, (5) Blink, and (6) Undefined, i.e., any sample that does not fit any of the other categories. Since GazeCom was collected at 250 Hz, we downsampled all fixations in the Lund dataset to 250 Hz.

## Description of the OK Lab dataset

The eye tracking database from our group is fully described in [60] and is labeled "GazeBase". All details regarding the overall design of the database, subject recruitment, tasks and stimuli descriptions, calibration efforts, and eye-tracking equipment are presented there. There were 9 temporally distinct "rounds" over a period of 37 months, and round 1 had the largest sample. This report only includes 19 subjects. We started with 20 but dropped one very low-quality recording. The subjects were randomly chosen from 322 subjects available from round 1. Briefly, subjects were initially recruited from the undergraduate student population at Texas State University through email and targeted in-class announcements. Subjects completed two sessions of recording (median 19 min. apart) for each round of collection. Each session consisted of multiple tasks. The only task employed in the present evaluation was the text reading task. This smaller dataset of 19 subjects recorded while reading the text we refer to as the "OK Lab" dataset. Each subject was asked to read, silently, an identical pair of quatrains from the famous nonsense poem, "Hunting for a Snark", written by Lewis Carroll (written from 1874-1876). The text was displayed in Times New Roman 20 pt. size bold font and was single-

spaced. The mean letter interval for each piece of text was approximately 0.50 degrees of the visual angle. The height of the line of the text was 0.92 degrees of visual angle. Monocular (left) eye movements were captured at a 1,000 Hz sampling rate using an EyeLink 1000 eye tracker (SR Research, Ottawa, Ontario, Canada).

The initial purpose of the OK Lab dataset was a study of inter-rater agreement [55]. In that study, fixations were coded as "1", saccades as "2", and PSOs as "3". All other classified events (blinks, unclassified data, forms of noise, and artifacts) were also coded. There were 4 raters. A very detailed set of manual classification rules was created to guide the raters. This manual is available at https://digital.library.txstate.edu/handle/10877/13373. Rater 1 was the best-performing rater, and so, in this paper, we only use the classifications made by rater 1.

Since the dataset was collected at 1000 Hz, we downsampled the fixation signals to 250 Hz to be more comparable to GazeCom.

For all entropy illustration figures, the data points represent samples from a time series, spaced 2.5 ms apart. Additionally, the horizontal position is measured in degrees of visual angle.

## Fixations durations for the three datasets

Initially, the fixation durations for the three datasets were statistically significantly different (see Fig 1(A)). The GazeCom dataset had 35,650 fixations, the Lund dataset had 409 fixations and the OK Lab dataset had 1,852 fixations. Before comparing the datasets on entropy measures we wanted to equalize the distributions of fixation durations across the datasets. The first step was to remove all fixations below 20 samples and above 80 samples. To further equalize the distributions, we randomly removed some higher duration fixations for both the GazeCom and the Lund datasets. The final duration violin plots are presented in Fig 1(B). Note that the medians are now very close and that the distributions are no longer statistically significantly different. After equalization, the GazeCom dataset had 20,613 fixations, the Lund dataset had 145 fixations and the OK Lab dataset had 874 fixations.

## Description of software used for calculations

For this report we used the toolkit "EntropyHub" (Please visit the website www.EntropyHub. xyz to see the guidelines for using *EntropyHub*.) that provides a wide range of functions to calculate different entropy statistics. *EntropyHub* is licensed under the Apache License (Version 2.0) and is free to use by all [61].

We originally evaluated 15 different entropy metrics in a pilot study. Based on the intercorrelations among the 15 metrics and the apparent relevance of each metric for discriminating between fixations, we chose six relatively uncorrelated and informative metrics for the formal study. In Table 2 we present the 6 entropy measures we chose for our analysis.

**Table 2. List of all entropy measures used.**

|   | Name | Abbrev. | $\tau$ * | $m$ * |
|---|------|---------|---------|-------|
| 1 | Fuzzy Entropy | FuzzEn | 1 | 1, 2 |
| 2 | Increment Entropy | IncrEn | 1 | 2 |
| 3 | Sample Entropy | SampEn | 1 | 1, 2 |
| 4 | Gridded Distribution Entropy | GridEn | - | - |
| 5 | Phase Entropy | PhasEn | - | - |
| 6 | Spectral Entropy | SpecEn | - | - |

*—phase space parameters

**Table 3. Key math symbols used.**

| Symbol | Meaning |
|---|---|
| $N$ | length of original time series |
| $L$ | number of states (rows) in phase space |
| $\tau$ | lag used for phase space reconstruction |
| $m$ | embedding dimension used for phase space reconstruction |
| $R^m$ | The universe of real number in dimension m |
| $r$ | distance threshold |
| $std$ | standard deviation |
| $\#N$ | number of something |

In Table 3 we present a list of math symbols that will be used throughout the paper. We also include some information about phase space parameters that will be explained below.

## Phase space reconstruction

**Introduction to phase space.** Since the three features described below employ the concept of phase space, this concept will be described here. Generally, the one-dimensional time series $X$ with a sampling rate $T$ given by the biological system can be mathematically represented as a system of differential equations [62]. Some components of the eye movement signal have been modeled at this time [63, 64]. However, no set of such equations exists to model the full complexity of entire eye movement signals.

By applying specific phase space reconstruction techniques from embedding theory [31], it is possible to reconstruct the time series into a high dimensional vector space and evaluate the complexity of the signal. For the reconstruction of the original system dynamics, we need the two embedding parameters [65]: $\tau$ time lag and $m$, the embedding dimension.

**Embedding parameters for the phase space reconstruction.** The initial time series $X$, with measurements equally spaced in time $x(1)$, $x(2)$, ..., $x(N)$, can be reconstructed into a 2-dimensional matrix ($Y$) with rows representing states of the system and columns representing the coordinates of these states. In other words, the initial signal $X$ is said to be embedded in a new matrix $Y$ such that $f(X) = Y$.

**Time lag, $\tau$.** The embedding parameter $\tau$ [66] is a period of time between two observations (samples) $x(i)$ and $x(j)$ such that $j > i$. The general formula for the time series $X$ given $\tau$ is:

$$X_\tau = \{x(1), x(1 + \tau), x(1 + 2 \cdot \tau), \ldots\} \tag{1}$$

For example, if $\tau = 4$, then:

$$X_{\tau=4} = \{x(1), x(5), x(9), \ldots\} \tag{2}$$

With this parameter, we can process a set of time series with different durations between samples.

**Embedding dimension, $m$,.** The embedding parameter $m$ [67] is the dimension of the space for the reconstruction of a phase portrait of $X$, such that $X$ in $Y \in R^m$, where $R^m$ is a m-dimensional euclidean space. Accordingly, we formally define our procedure as $f: X \to Y \in R^m$.

**Phase space construction.** We construct a set $Y$ of probable state variables from the initial signal $X$. $Y$ is a matrix of possible states. Every possible state $y(i) \in Y$ is represented by a delayed

measure of *X*. Function $f: X \rightarrow Y$ is called a delay embedding procedure. As noted above, the two input parameters of *f* are embedding parameters $(\tau, m)$.

There are many potentially different phase space reconstructions (*Y*) for a given *X* time series, each characterized by different embedding parameters [68]. The *X*-sequence of length *N* will be divided into the sequence of vectors *Y* of length *L* where the value of *L* can be defined as $L = N - (m-1)\tau$. In other words, if the signal time series has a length of 23, *m* = 3, and $\tau = 4$, the length *L* of *Y* will be 15.

Finally, *Y* will be in the next form:

$$Y = \{y(i)\}, \quad i = \overline{1, M}, \tag{3}$$

$$y(i) = x(i), \ x(i + \tau) \ , \ldots, \ x(i + (m-1) \cdot \tau). \tag{4}$$

where each vector *y*(*i*) has length *m* and time difference between elements in vector *y*(*i*) is $\tau$.

In our analysis, described in detail below, we create one phase space per fixation with parameters $\tau = 1$, *m* = 2.

**Illustrate phase space reconstruction.** To further illustrate phase space reconstruction, we present Tables 4 and 5. Table 4 contains an actual 23 sample fixation (horizontal position). We employ the MATLAB routine "phaseSpaceReconstruction" which takes as input the signal and produces the optimized estimated $\tau$ ($\tau = 4$ in this case), using average mutual information [69]), the optimized embedding dimension *m* (*m* = 3 in this case, using the False Nearest Neighbor algorithm [70], and the phase space reconstruction in Table 5. Note that a number that is the 3rd coordinate of the 1st state is identical to the 2nd coordinate of the 5th state

**Table 4. Raw signal.**

| Row | Signal |
|---|---|
| 1 | -3.2406 |
| 2 | -3.2393 |
| 3 | -3.2428 |
| 4 | -3.2439 |
| 5 | -3.2388 |
| 6 | -3.2241 |
| 7 | -3.2065 |
| 8 | -3.191 |
| 9 | -3.1842 |
| 10 | -3.1828 |
| 11 | -3.1846 |
| 12 | -3.1857 |
| 13 | -3.1871 |
| 14 | -3.1869 |
| 15 | 3.1871 |
| 16 | -3.1841 |
| 17 | -3.176 |
| 18 | -3.1617 |
| 19 | -3.144 |
| 20 | -3.128 |
| 21 | -3.1186 |
| 22 | -3.1171 |
| 23 | -3.1231 |

**Table 5. Phase space reconstruction for the signal in Table 4 to the left.**

| State | First Coordinate | Second Coordinate | Third Coordinate |
|---|---|---|---|
| 1 | -3.2406 | -3.2388 | -3.1842 |
| 2 | -3.2393 | -3.2241 | -3.1828 |
| 3 | -3.2428 | -3.2065 | -3.1846 |
| 4 | -3.2439 | -3.1910 | -3.1857 |
| 5 | -3.2388 | -3.1842 | -3.1871 |
| 6 | -3.2241 | -3.1828 | -3.1869 |
| 7 | -3.2065 | -3.1846 | -3.1871 |
| 8 | -3.1910 | -3.1857 | -3.1841 |
| 9 | -3.1842 | -3.1871 | -3.1760 |
| 10 | -3.1828 | -3.1869 | -3.1617 |
| 11 | -3.1846 | -3.1871 | -3.1440 |
| 12 | -3.1857 | -3.1841 | -3.1280 |
| 13 | -3.1871 | -3.1760 | -3.1186 |
| 14 | -3.1869 | -3.1617 | -3.1171 |
| 15 | 3.1871 | -3.1440 | -3.1231 |

which is itself identical to the 1th coordinate of the 9th. These differences in position represent the effect of $\tau = 4$, the time lag.

In Fig 2, we plot the phase space (in Table 5) for this particular fixation period (in Table 4).

## Entropy-related features

**Shannon Entropy: A fundamental measure employed in various entropy algorithms.** By definition, "Shannon Entropy" (SE) measures the amount of information $H(X)$ contained in a continuous time series of events $X$ [71]. If our time series is simple, it will be characterized by low entropy; whereas, if it is complex, it will exhibit high entropy. For the discrete random variable $X$ with possible outcomes $x_1, x_2, \ldots, x_N$ we create, a series $Y$ where every state $y_m(i)$ is presented as a pattern with $m$ samples. We first need to determine if each state is unique or if some states are repeated, and associate every unique state with the probability that describes its occurrence in the phase space. Once we have a set of probability values, $SE$ can be formally defined as:

$$SE(m) = -\sum_m P_m \log P_m,$$  (5)

where $P_m$ stands for the joint probability of the pattern $y_m(i)$ and the sum is extended to all the different $y_m(i)$ patterns [72].

**Entropy features that require phase space reconstruction.** *Metric 1: Fuzzy entropy.* In 1965, Zahed [73] introduced the mathematical concept of a "fuzzy set". A fuzzy set characterizes the input-output relationship in an environment of imprecision. Fuzzy Entropy(FuzzEn) is based on this concept. As explained below, the fuzzy function produces a measure of the similarity between the two states, based on their shape (trajectory) in phase space.

For the $N$-sample series $X$ we apply the phase space reconstruction technique with $(\tau, m)$. We generalize every state as follows:

$$y(i) = \{x(i), x(i+1), \ldots, x(i+m-1)\} - x_0(i),$$  (6)

**Reconstructed Fixation in 3-dimensional Phase Space**

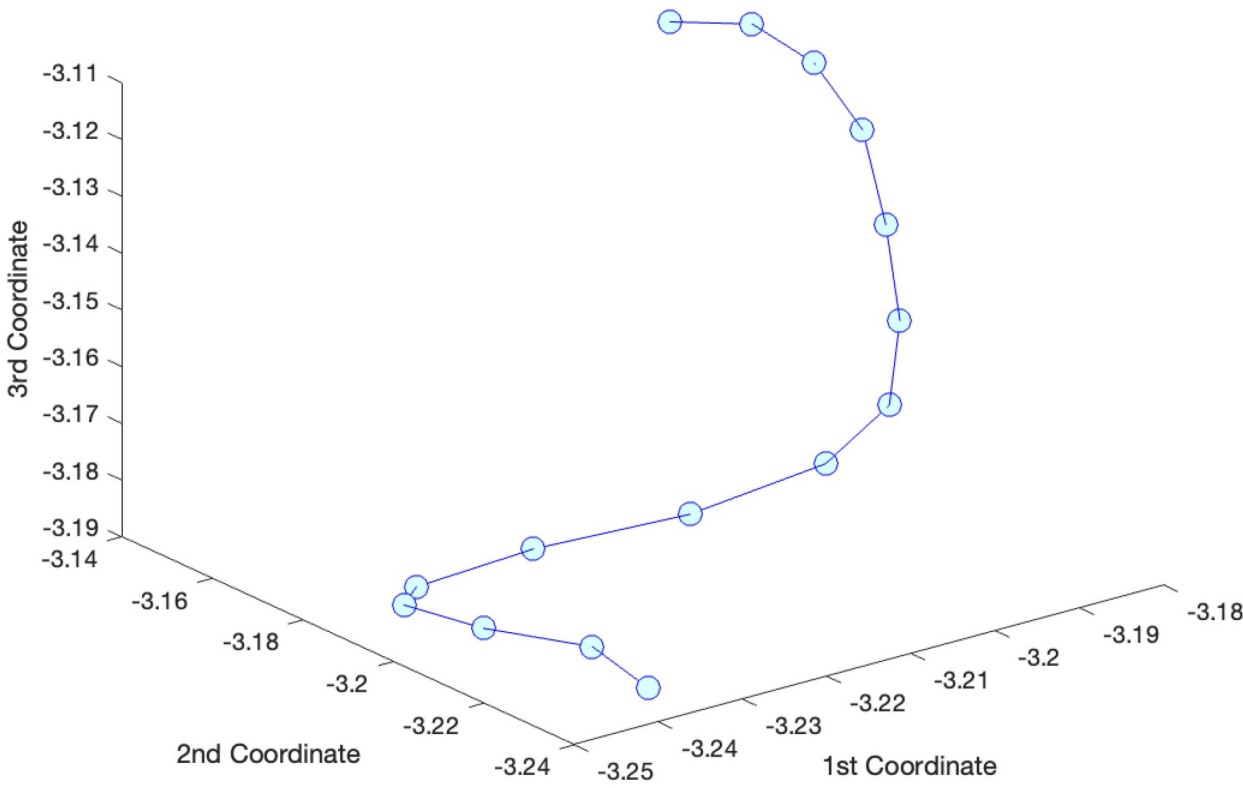

**Fig 2. Plot of the trajectory of the states thru the phase space from Table 5.**

where *X*-values are generalized by subtracting the mean of the signal [74]:

$$x_0(i) = \frac{1}{m} \sum_{j=0}^{m-1} x(i+j). \tag{7}$$

For a given vector $y(i)$, we define the distance between vector $y(i)$ and another vector $y(j)$ as the maximum absolute difference of the corresponding scalar components (coordinates) [74]:

$$\begin{aligned} d_{ij}^m &= d[y(i), y(j)] \\ &= \max_{k \in (0, m-1)} |(x(i+k) - x_0(i)) - (x(j+k) - x_0(j))|. \end{aligned} \tag{8}$$

To evaluate the relationship between two vectors $y(i)$ and $y(j)$, we use the concept of "membership degree" from [73] with a fuzzy function $\mu$ [74] that associates each point $y(i)$ with a real number in the interval [0, 1]. Following Zadeh's work [73], we estimate the degree of similarity for a particular state $y(i)$. The nearer the value of the fuzzy function is to unity, the higher the degree of similarity. Using *n* and *r* values, where n is a gradient of the boundary of the exponential function (estimated by the algorithm), we calculate the "membership degree" $D_{ij}$

of $y(i)$ to $y(j)$ through a fuzzy function $\mu(d_{ij}^m, n, r)$ as:

$$D_{ij}^m(n, r) = \mu(d_{ij}^m, n, r),$$

$$\mu(d_{ij}^m, n, r) = \exp(-\frac{(d_{ij}^m)^n}{r}). \tag{9}$$

Then we create the variable $M = N - m$ and define the function $\phi^m$ for the two embedding dimensions equal to $m$ and $m + 1$:

$$\phi^m(n, r) = \frac{1}{M} \sum_{i=1}^{M} (\frac{1}{M-1} \sum_{j=1, j \neq i}^{M-1} D_{ij}^m) \ , \tag{10}$$

$$\phi^{m+1}(n, r) = \frac{1}{M+1} \sum_{i=1}^{M+1} (\frac{1}{M} \sum_{j=1, j \neq i}^{M} D_{ij}^{m+1}). \tag{11}$$

Finally, using Eqs (10) and (11), we determine the FuzzEn [74] of the finite dataset $X$ as the negative natural logarithm of the deviation $\phi^m$ from $\phi^{m+1}$:

$$FuzzEn(m, n, r) = \ln \phi^m(n, r) - \ln \phi^{m+1}(n, r). \tag{12}$$

*Illustration example: Fuzzy entropy.* In this subsection, we illustrate the FuzzEn for 3 fixations, one with low FuzzEn, one with intermediate FuzzEn, and one with high FuzzEn. These plots all employ $m = 1$. As a reminder, we need to construct two phase spaces with $m = 1$ and $m = 2$ to calculate FuzzEn(1, n, r).

FuzzEn employs "fuzzy functions" which assess the similarity, or "membership degree" of the states, based on their shape (trajectory) in phase space. The input to each fuzzy function is the distance between two states. If the distance between the states is high, the membership degree produced by the fuzzy function is low. We noticed that for the fixations without any relatively large state-to-state transitions, the membership degree will be high. However, when the fixation trajectory contains very large state-to-state transitions, the membership degree will be very small. This will be reflected in the final entropy calculation.

In Fig 3, we see that the FuzzEn is higher for the fixations in Fig 3A and 3C than for the fixation in Fig 3E. This is because the fixations in Fig 3A and 3C both contain brief but relatively large eye position changes. Whenever we have large position differences in a fixation, we will also have large state distances in the $m = 2$ phase space (Fig 3B and 3D). The fixation in Fig 3E does not exhibit any noticeably large eye-position transitions, resulting in a low value for FuzzEn.

*Metric 2: Increment entropy.* For the first step in the calculation of Increment entropy (IncrEn) [75], we use the time series $X$ of length $N$ to calculate the vector of sample-to-sample differences $\Upsilon$ of length $N - 1$. We refer to this vector of sample-to-sample differences as the "incremented X-sequence", $\Upsilon$:

$$\Upsilon = \{v(i), 1 \leq i \leq N - 1\}, \tag{13}$$

where $v(i) = x(i + 1) - x(i)$. Next, we create a phase space from the incremented X-sequence, $\Upsilon^m$, where $m \geq 1$. We characterize the sign of the change between the one coordinate and its neighbor coordinate as either -1, 0, or 1. We do this for each pair of adjacent coordinates. The

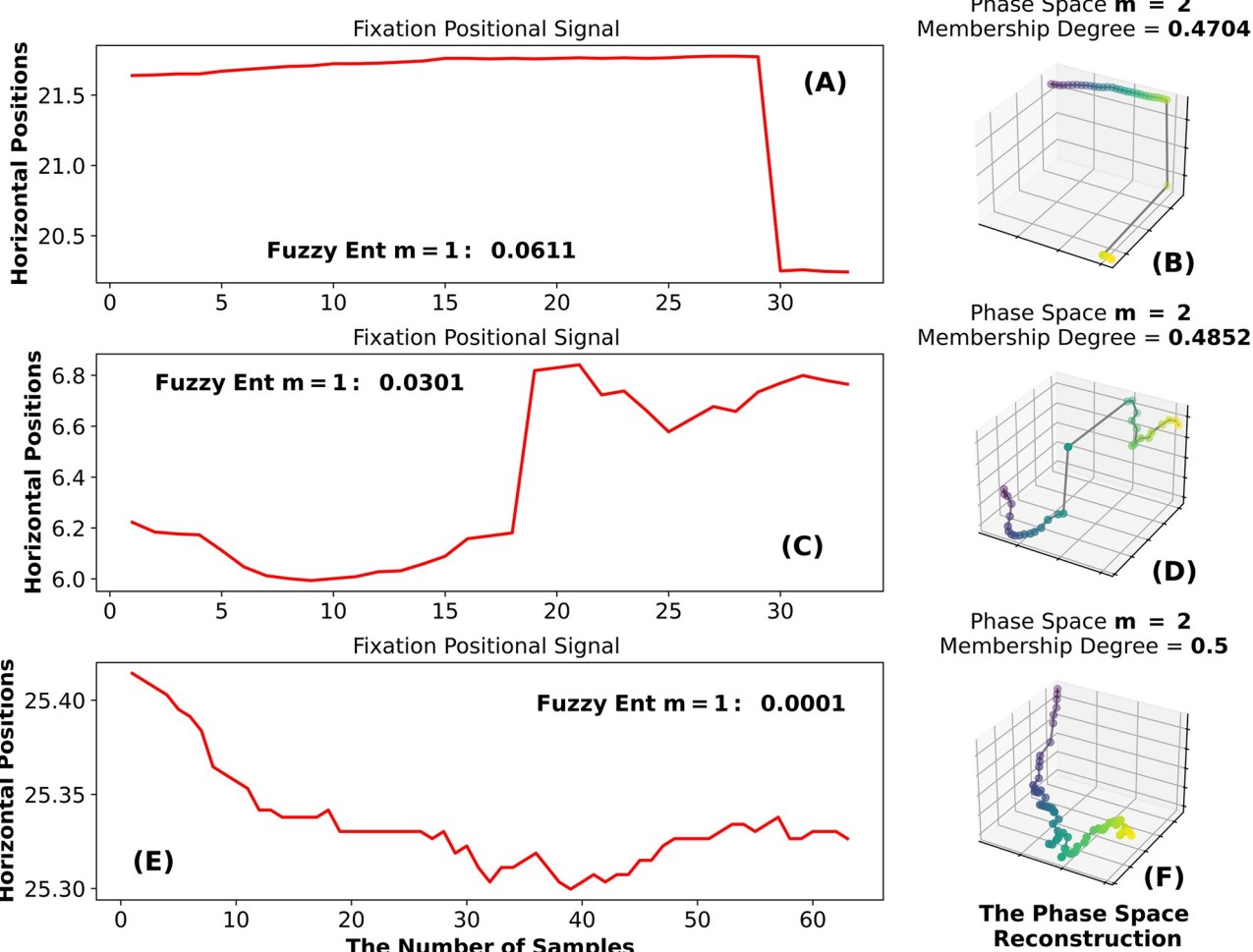

**Fig 3. Explanatory plot for the fuzzy entropy.** In Fig 3A we have a fixation with high position change and relatively high FuzzEn. In Fig 3B we see the phase space reconstruction with the parameters $\tau = 1$ and $m = 2$ for the fixation in Fig 3A and provide the membership degree for the trajectory in Fig 3B. In Fig 3C, we present a fixation with smaller position change and its phase space reconstruction in Fig 3D. In Fig 3E, we present a fixation with the lowest FuzzEn value and its 2-D trajectory in Fig 3F.

magnitude of the change between adjacent coordinates is calculated as follows:

$$q(j) = \begin{cases} 0, & std(\Upsilon^m) = 0 \\ min(qr, \frac{||v(j)|| \cdot qr}{std(\Upsilon^m)}), & std(\Upsilon^m) \neq 0, \end{cases} \tag{14}$$

where $qr$ is referred to as the quantifying resolution [75]. At this point, each change between adjacent coordinates for each state can be summarized by a 2-symbol code [sign, magnitude]. Each state can be represented as a series of symbol codes $(m - 1) \cdot 2$.

Next, we need to count the frequency of occurrence of each state's pattern. The probability of each state pattern is the frequency of occurrence of the state pattern divided by the length of the time series. Plugging these probabilities into SE Eq (5) we get IncrEn.

*Illustration example: Increment entropy.* In Fig 4 we have the chart for Increment Entropy (IncrEn). As we can see, the first 3 examples from the top row (Fig 4A to 4C) are very rough fixations with complex trajectories. In the middle row, the trajectories are smoother, but they

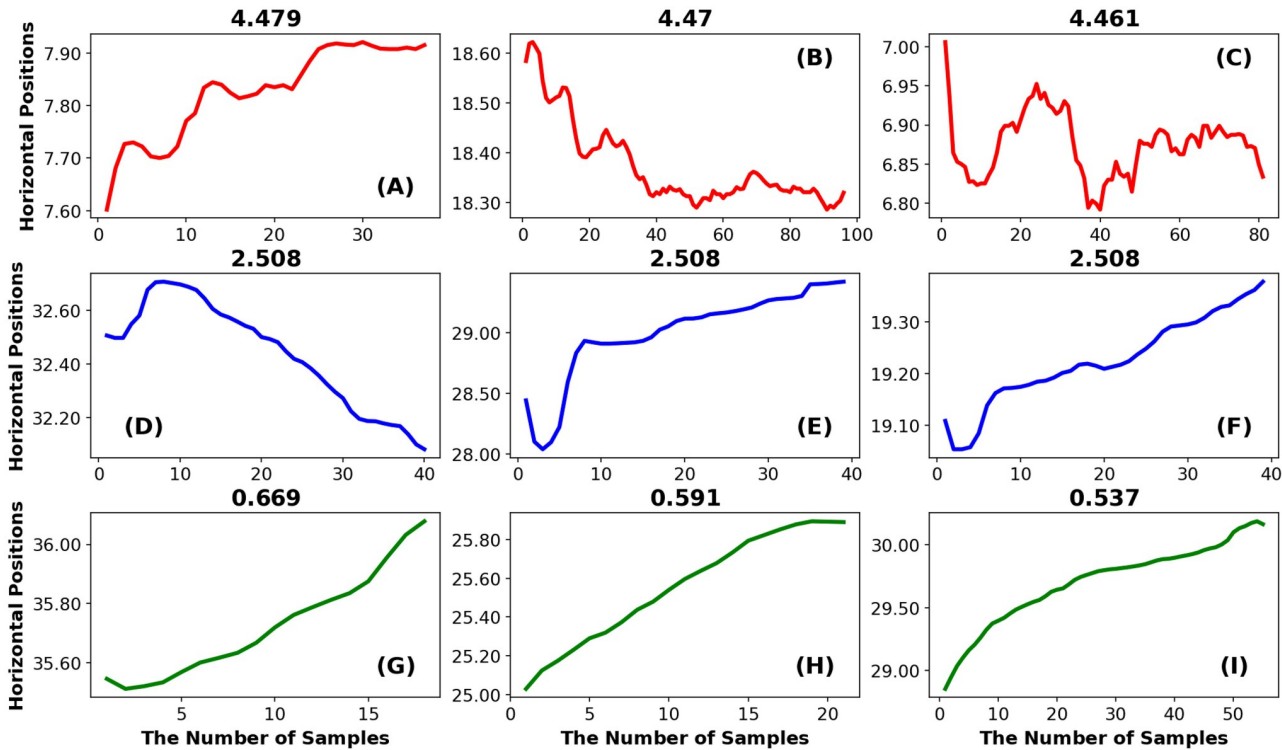

**Fig 4. The Increment Entropy.** The top row illustrates 3 fixations with high IncrEn, the middle row illustrates fixations with intermediate IncrEn and the bottom row illustrates fixations with low IncrEn.

are still somewhat complex with fewer marked peaks and valleys. For the bottom row, the fixation trajectories are smoothest without any noticeable valleys or peaks.

For the IncrEn algorithm, the sample-to-sample differences (referred to as the incremented time series) are used, and the phase space reconstruction is applied to this incremented time series. For every state, a "symbol pattern" is created by using the sign and the magnitude of change between the adjacent coordinates in each state. So, for fixations with many changes in sign and magnitude (see Fig 4A–4C), the IncrEn value is large. For fixations with fewer sign and magnitude changes the IncrEn is somewhat lower (see Fig 4D–4F). If there are few sign and magnitude changes, the IncrEn value is low (see Fig 4G–4I).

*Metric 3: Sample entropy.* Sample Entropy (SampEn) is considered an improvement over Approximate Entropy (ApEn) [76, 77]. The value of ApEn is dependent on the data length, i.e., the longer the length, the higher the ApEn. SampEn is more accurate and less dependent (although still dependent, see below) than ApEn on data length. For the SampEn, we use the same phase space reconstruction technique to rebuild a sequence $X$ into a sequence $Y^m$ in real m-space $R^m$. We also use the Chebyshev formula to calculate the distance $d(y^m(i), y^m(j)) = d_{ij}^m, i \neq j$ between every pair of $Y$-states. For the fixed parameters $m$ and $r$, we determine the two values $A$ and $B$ such that:

$$A = \#N\{(y^m(i), y^m(j))|d_{ij}^m \leq r\}, \tag{15}$$

$$B = \#N\{(y^{m+1}(i), y^{m+1}(j))|d_{ij}^{m+1} \leq r\}. \tag{16}$$

As a result, the SampEn will be determined as:

$$SampEn(m, r) = -\log\frac{A}{B}, \tag{17}$$

where the number $\frac{A}{B}$ can be considered as the conditional probability that two sequences within a tolerance $r$ for $m$ points remain within $r$ of each other at the next point [78].

*Illustration example: Sample entropy.* Sample Entropy uses phase space reconstruction. This produces a matrix whose rows are states and whose columns are coordinates. The number of columns is determined by the embedding dimension (m). We used the default value $m = 1$. In this case, each state consists of sequential positions in the raw signal and can be thought of as existing in a 1-dimensional space. For the computation of this entropy, we start with a list of distances, specifically Chebyshev distances, between each state in 1-dimensional space. Sample Entropy considers the Chebyshev distances between all possible states and calculates the number of pairs for which the distance is smaller than, or equal to the fixed radius distance $r$. We refer to this count of distances less than $r$ as $B$. In the default case, $r = 0.2 * std(X)$. The next step in the calculation of Sample Entropy is to create a new phase space using $m = 2$ (a 2-dimensional space). Once again, the distances between all possible pairs of states are computed and the number of distances less than or equal to $r$ is counted, we refer to this number as $A$. Sample Entropy is

$$SampEn = -\log\frac{A}{B}, \tag{18}$$

In Fig 5, we present a chart to help explain Sample Entropy. We have chosen 3 fixations with very similar threshold r values. This also means that we have chosen 3 fixations with very similar STDs. In Fig 5A, we present a very rough fixation in which a small percentage of the distances are less than r. In Fig 5B we present a 2-D fixation trajectory in which a very small number of distances are less than r. In the case of Fig 5A and 5B, the ratio of $A/B$ in percent terms is 0.30. This produces a sample entropy of 1.26. In Fig 5C we present a smoother fixation (at least smoother than Fig 5A) with a larger percentage of distances less than r. In Fig 5D, for the 2-dimensional trajectory, a somewhat larger percent of distances is less than r. For this pair (Fig 5C and 5D), the ratio of $A/B$ is 0.50, a somewhat larger ratio than for Fig 5A and 5B. This leads to Sample Entropy of 0.73, which is 57% less than the entropy for Fig 5A and 5B pair. In Fig 5E, we present a fixation that, with the exception of the saccade-like portion at the start, is very smooth. Not surprisingly, a very large percent of distances were less than a threshold. In Fig 5F, the percent of distances less than the threshold ($r$) is also relatively large. In this case, the ratio of $A/B$ is almost 1.0, substantially larger than the other pairs. This is associated with a Sample Entropy of 0.26.

**0.0.1 Entropy features that do not require phase space reconstruction.** *Metric 4: Gridded Distribution Entropy.* Initially, with a constant sampling rate of 250 Hz, $X$ is normalized within the range [0, 1] by a min-max normalization [79]:

$$x_{norm}(i) = \frac{x(i) - \min(X)}{\max(X) - \min(X)}. \tag{19}$$

Next, the gridded Poincaré plot [80] is constructed for $X_{norm}$. For each potential $n$ horizontal grid lines and $n$ vertical grid lines, the Gridded Distribution Rate (GDR) is an index of the goodness of fit of the grid. The Gridded Distribution Rate (GDR) is the proportion of all cells

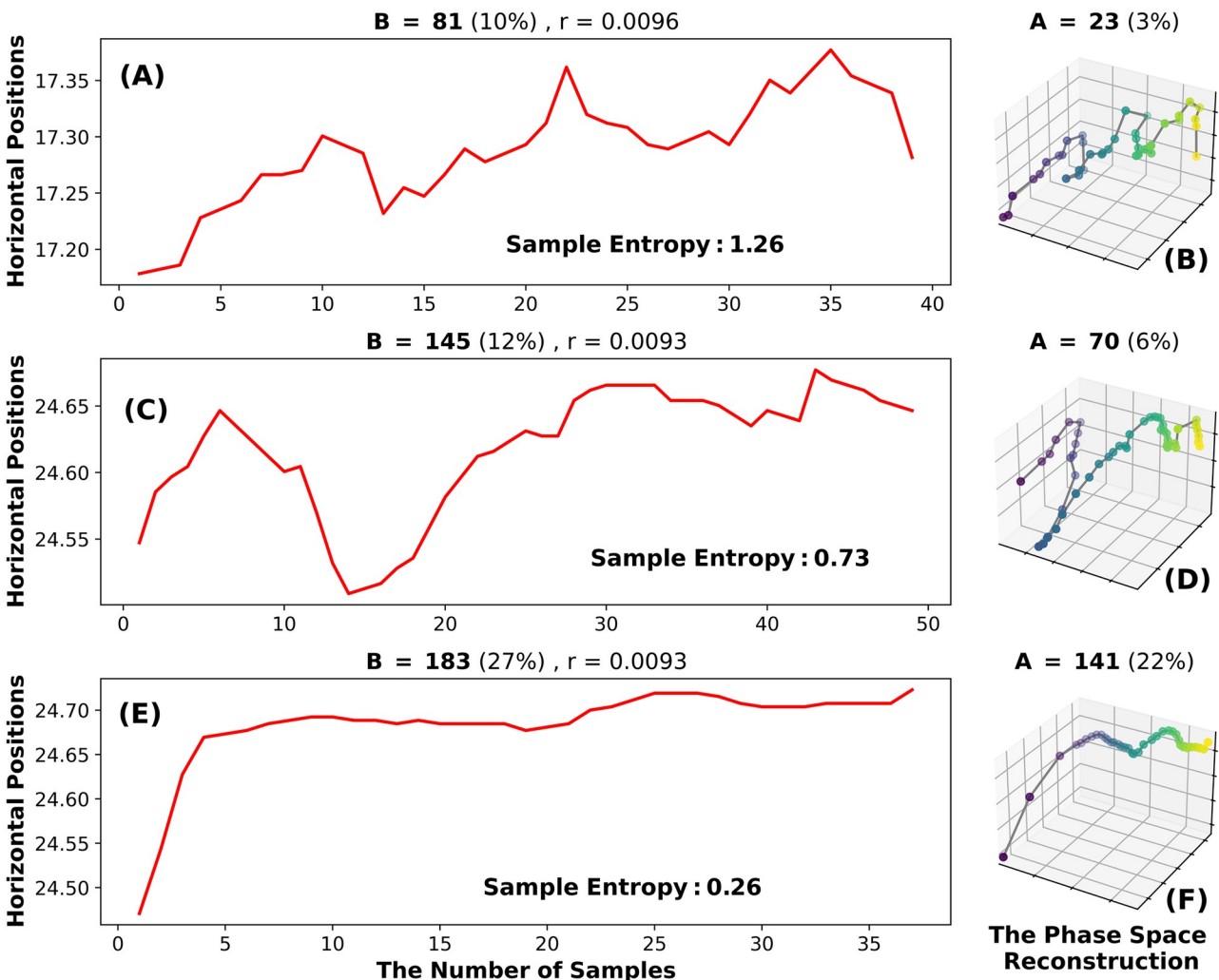

**Fig 5. Explanatory plot for the Sample Entropy.** In Fig 5A we have a relatively chaotic fixation. In Fig 5B we see the phase space reconstruction with the parameters $\tau = 1$ and $m = 2$ for the fixation in Fig 5A. In Fig 5C, we present a much less chaotic fixation and its 2-D fixation trajectory Fig 5D. In Fig 5E, we present a fixation with the lowest *SampEn* value and its reconstructed trajectory Fig 5F.

$(n^2)$ that are filled with at least one point $(\alpha)$ [81]:

$$GDR = \frac{\alpha}{n^2} \tag{20}$$

Next, we define the probability $P_j$ which specifies the frequency of points for the $j$-th grid in the gridded Poincaré plot:

$$P_j = \frac{b}{N-1}, \tag{21}$$

where $b$ is the number of points in the grid [81]. And finally, we use the SE formula Eq (5) to

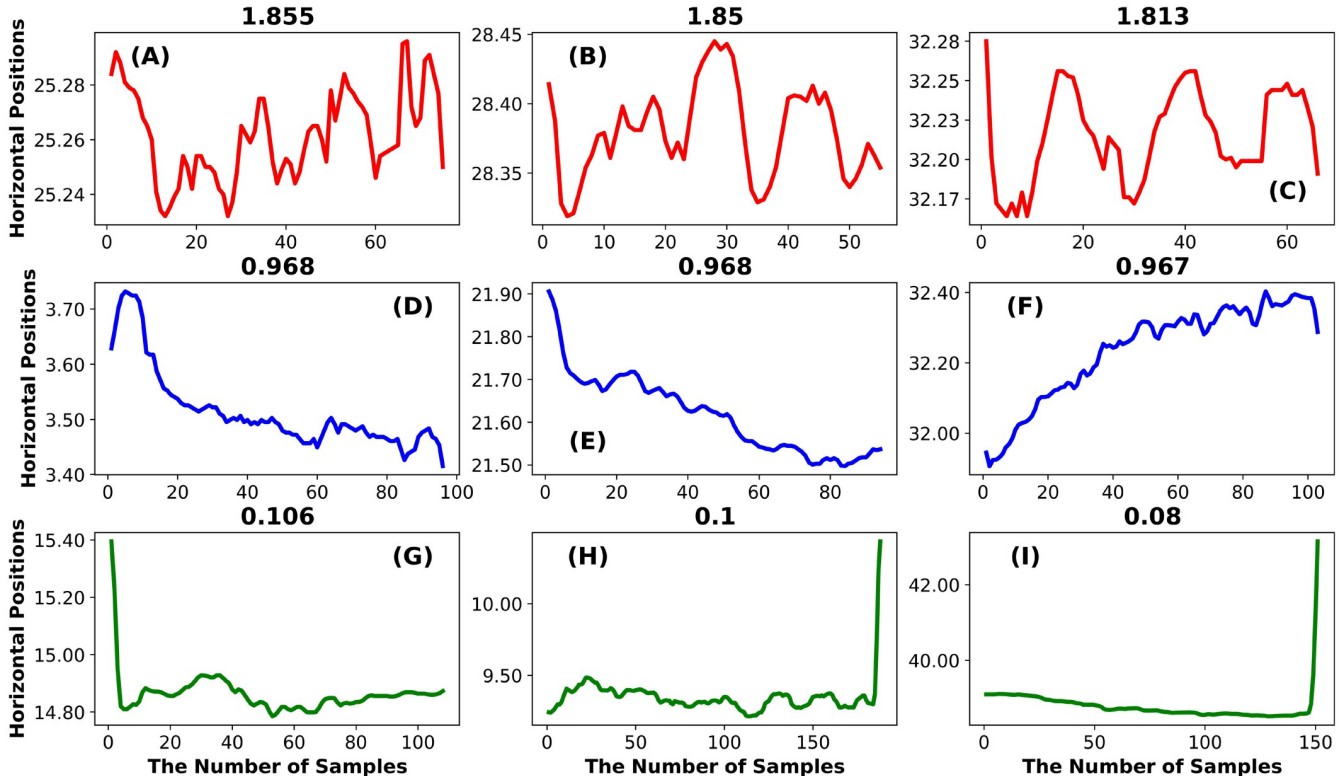

**Fig 6. The gridded Distribution Entropy.** In Fig 6A–6C there are three examples of fixation when this entropy is high. In Fig 6G–6I there are three examples of fixation when this entropy is low. In Fig 6D–6F there are three examples of fixations with entropy values intermediate between the high values and low values.

compute the Gridded Distribution Entropy (GridEn) calculation:

$$GridEn = -\sum_{j=1}^{n^2} P_j \log P_j. \tag{22}$$

*Illustration example: Gridded Distribution Entropy.* Fig 6 illustrates 9 example fixations and associated GridEn. As we can see, the top row examples have high GridEn (see Fig 6A to 6C). Note that these are rough fixations with lots of irregular chaotic transitions and local minimums (or maximums). In the middle row, the trajectories are smoother without sharp unexpected transitions, but they are still somewhat complex with fewer marked peaks and valleys. For the bottom row, the inclusion of corrective saccades at the start of the plot in Fig 6G and the inclusion of part of a saccade following each fixation in Fig 6H and 6I are noteworthy. The saccade dominates the signal and sets the amplitude range for fixations to be large. Relative to this large range, the non-saccade associated signal has a very small range. This leads to a low GridEn entropy for these events.

GridEn is a feature that does not require phase space reconstruction. It utilizes the Poincare plot of the fixation trajectory to study the distribution of sample position across *n* separate regions. In the illustrative example (see Fig 7), the sample positions on each axis are scaled to a range of 0 to 1. The plot is then divided into nine square regions, and the number of points within each region is counted. A time series is considered complex when its points are highly distributed across all regions on the Poincare plot (see Fig 7A–7C). Typically, noisy fixation trajectories will be scored by a high value of GridEn. On the other hand, time series that

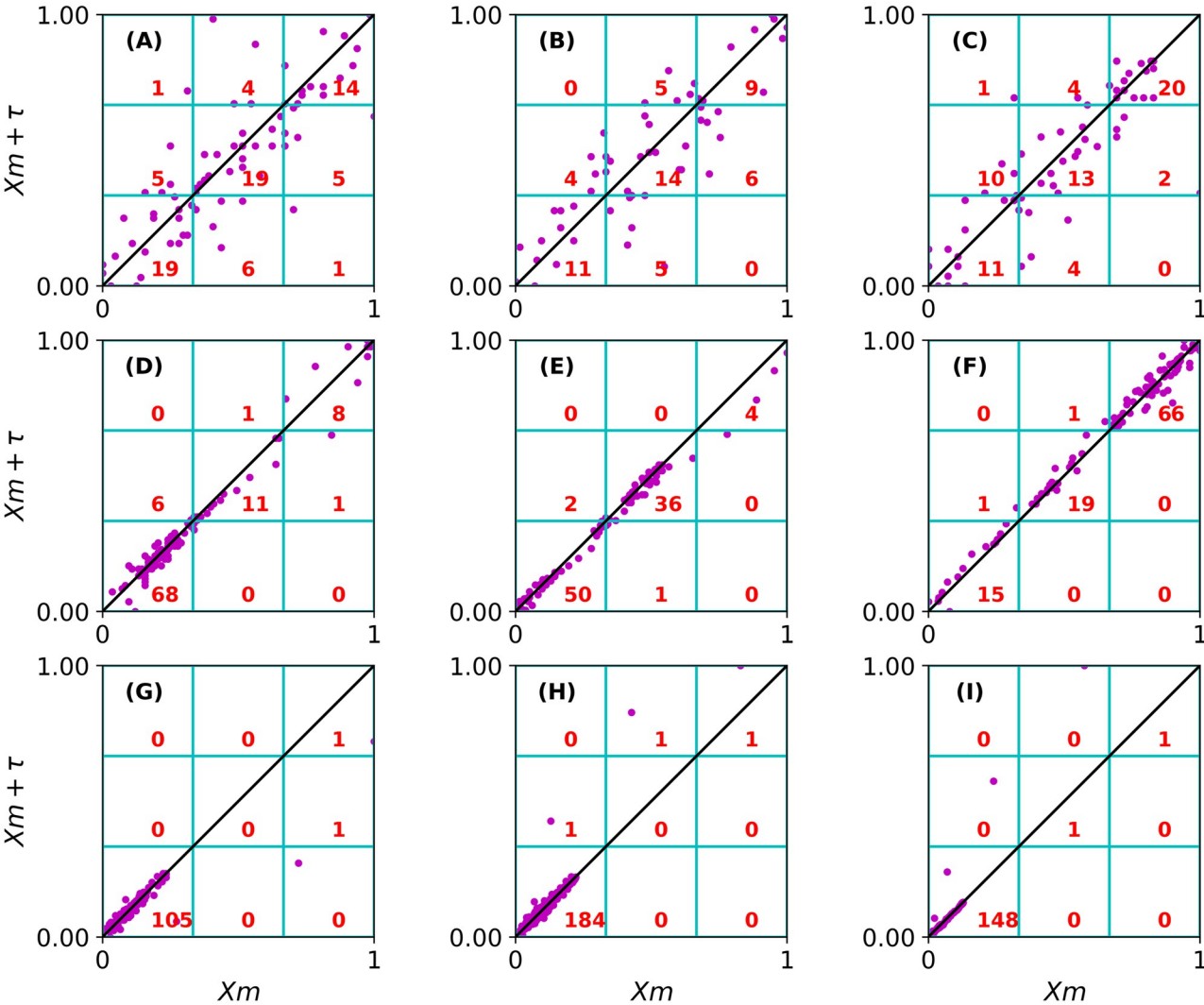

**Fig 7. The Poincare plot for the GridEn for each fixation is illustrated in Fig 6.** Each Poincare plot is divided into a 3-by-3 grid (9 squares), corresponding to the fixation with the same letter in Fig 6. In the right corner of each square, we present the number of points in that region. Fig 7A–7C are examples of fixations that exhibit a high value of GridEn because points on the Poincare plot are highly distributed across all nine regions. When points on the Poincare plots are mostly distributed across 2 (or 3) dominant regions, the corresponding trajectories will be characterized by an average value of GridEn (see Fig 7D–7F). In Fig 7G–7I there are three examples of fixations in which points are mainly located in one dominant region. This produces a low GridEn.

incorporate fixation and corrective saccade will be scored by a low value of GridEn (see Fig 7G–7I). This is because the samples corresponding to the corrective saccade will appear as noticeably distinct points on the Poincare plot.

*Metric 5: Phase entropy.* The Phase Entropy (PhasEn) analysis [82] begins with a second-order difference plot (SODP) [83]. Let consider a data sequence $X = x_0, x_1, x_2, \ldots, x_n$. From the given time series $X = \{x_n\}$, we will denote the two sequences of differences as $Y$ and $Z$:

$$Y = x_{n+2} - x_{n+1}, \tag{23}$$

$$Z = x_{n+1} - x_n. \tag{24}$$

The SODP is a plot of Y against Z. The slope from each line from the origin (0,0) to each point is determined. For this purpose, a four-quadrant arctan function is used to get the span of slope angles ∠ in the range $(0, 2\pi)$:

$$\theta = \tan^{-1} \frac{Y}{Z}. \tag{25}$$

The scatter plot can be divided into a variable number of angular or radial sectors. In our case, we used the default value of $k = 4$. These 4 sectors have an angle span of 90 degrees ($\frac{2\pi}{4}$ radians) each. The parameter $M_i$ where $i = \overline{1, k}$ will denote the number of points in $i$-th sector. For each sector, we will calculate the cumulative slope $S_\theta$:

$$S_\theta(i) = \sum_{j=1}^{M_i} \theta_j. \tag{26}$$

Next, we denote the cumulative slope of the entire plane as $S$, and divide each value of $S_\theta$ by it to obtain the probability distribution $P_i$ for every sector:

$$P_i = \frac{S_\theta(i)}{S}, \quad where \ S = \sum_{i=1}^{k} S_\theta(i). \tag{27}$$

As a result, using the SE formula (5), we get the PhasEn:

$$PhasEn = \frac{-1}{\log k} \sum_{i=1}^{k} P_i \log P_i. \tag{28}$$

*Illustration example: Phase entropy.* In Fig 8, we illustrate 9 fixations with different levels of PhasEn. In Fig 9, we present SODP plots for each fixation in Fig 8.

The SODP plot can be divided into any number of regions but typically the default (which we use) is to divide the space into 4 quadrants (Fig 9). We can see that the first 3 fixations examples from Fig 9A to 9C begin with a large signal change in the rightward direction. These large signal changes produce points in the SODP clustered in the upper-right quadrant. All these fixations from Fig 9A to 9C are chaotic with a lot of transitions. After these initial large signal changes the changes in the remaining fixation points will tend to be clustered around the origin in the SODP plot, with some points in all sectors. And such dispersed distribution of points across sectors will produce a large Phase Entropy.

Now let's look at the bottom row of Fig 8. Plots Fig 8G–8I exhibit a strong trend (drift) in one direction (left). This produces an SODP with values clustered in the lower-left (Fig 9). This uneven distribution of points over quadrants will produce a low Phase Entropy value. This also applies, to a smaller extent, to fixations in the middle row.

*Metric 6: Spectral entropy.* The Spectral Entropy (SpecEn) is calculated based on a Fast Fourier Transforms (FFT) of the signal [84]. Every motion is characterized by a countable number of frequency components. And generally, there are a few of the frequency components that are dominant.

Every frequency component $v_r$ of the signal $X$ can be associated with the probability $P_r$ such that:

$$P_r = \frac{|f_r|^2}{\sum_{r'} |f_{r'}|^2}. \tag{29}$$

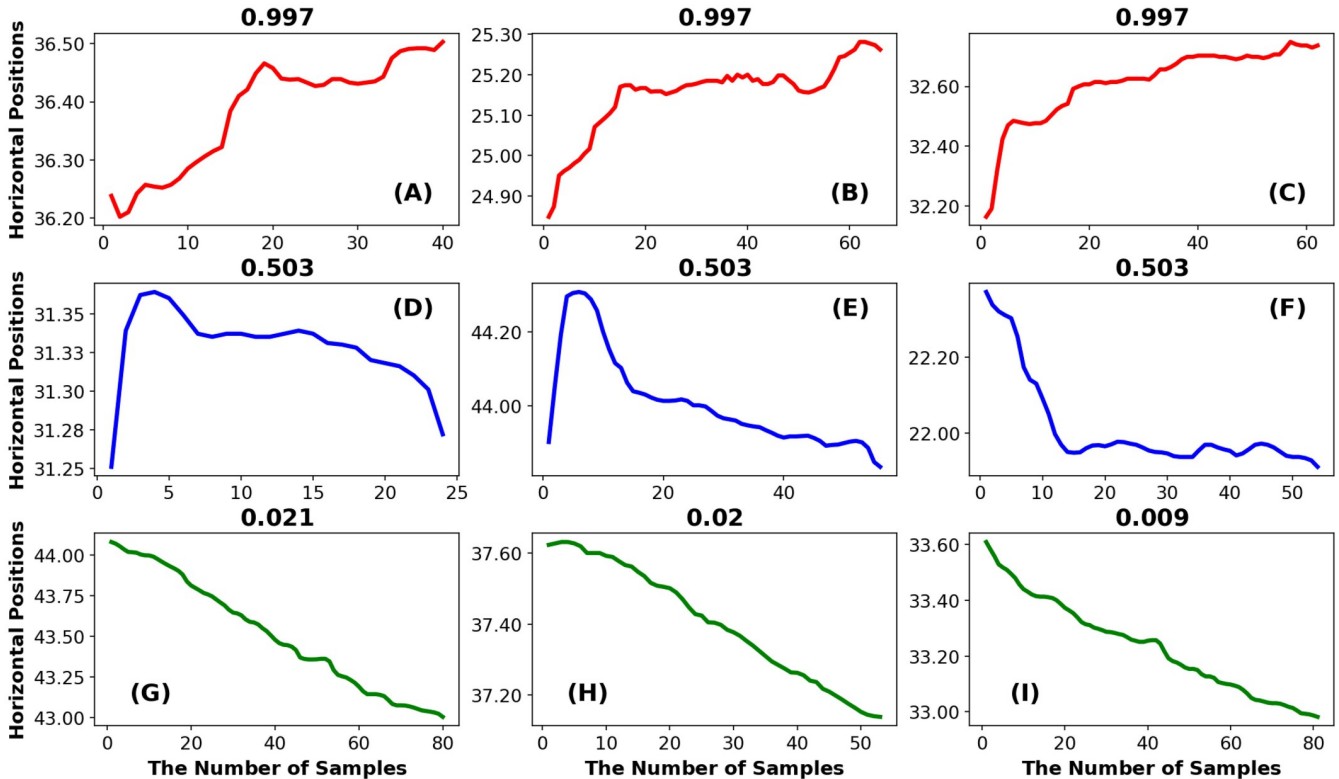

**Fig 8. The Phase Entropy.** In this plot, we present the 9 fixations. The top row consists of examples with high PhasEn, the bottom row consists of examples with low PhasEn and the middle row consists of examples with intermediate PhasEn.

As a result, by using the SE formula (5), we can get the SpecEn in the next form:

$$SpecEn = -\sum_r P_r \ln P_r. \tag{30}$$

*Illustration example: Spectral entropy.* SpecEn is calculated based on a spectral density analysis of the eye-tracking signal. This analysis produces spectral density estimates for all possible frequencies for a given signal. The number of frequencies is a function of the length of the fixation. The spectral density estimates for all frequencies are normalized by the sum of the power spectral density across all frequencies. The Spectral Entropy is the result of computing the *SE* of these relative spectral density estimates.

In Fig 10 we present three fixations: one with high SpecEn Fig 10A, one with medium SpecEn Fig 10C and one with low SpecEn Fig 10E. The relative spectral density for each frequency is presented in Fig 10B–10F. Note that we have rescaled the ordinate to range from 0.0 to 0.001. This scaling allows us to visualize the majority of frequencies for which data are available. The fixation in Fig 10A appears relatively chaotic compared to those in Fig 10C and 10E. It produces a spectral density plot with many different relative density estimates which leads to a higher value of SpecEn. The fixation trajectory in Fig 10E is the smoothest, and therefore one would expect very low and consistent normalized spectral density in the higher frequencies. Since there are higher frequencies than low frequencies, the consistent normalized density estimates in Fig 10F result in a low estimate of SpecEn.

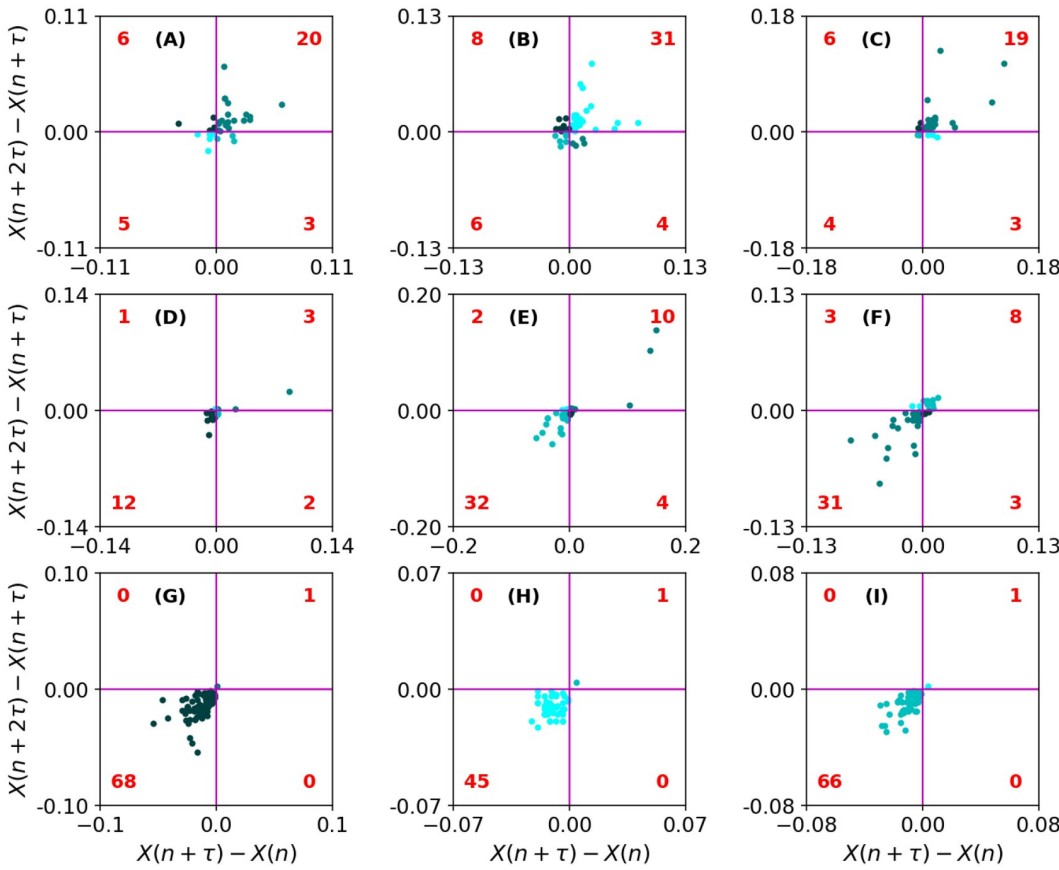

**Fig 9. The Second Order Differences Plots (SODP) for each fixation illustrated in Fig 8.** The top row shows SODP plots for the 3 fixations previously illustrated with high PhasEn. The middle row displays 3 fixations with intermediate PhasEn, and the bottom row depicts 3 fixations with low PhasEn. For each fixation, the SODP is divided into 4 radial sectors. In the corner of each sector, we display the number of points contained within that sector. Fig 9A to.9C depict three examples of fixation where the SODP points are highly distributed across the four sectors. Fig 9D to.9F present three examples of fixation where the SODP points are less randomly distributed across the four sectors compared to the first three examples, but more randomly distributed than the last three examples. Finally, Fig 9G to.9I display three examples of fixation where the SODP points are predominantly located within a single dominant sector.

## High-low analysis

To describe what kinds of fixations have very high and very low entropy measures, we have performed what we call a High-Low Analysis. We created a total of 36 figures by utilizing three datasets (GazeCom, Lund and OK Lab), six final entropy measures, and two figures per entropy measure (representing high entropy and low entropy, see Figs 11 and 12). The fixations are presented in the context of 30 samples before the fixation and 30 samples after the fixation.

The three authors of this report each rated each fixation to decide if it was a "good" fixation or if it was a misclassified one (see Table 7). We considered six types of misclassifications as detailed in Table 6. We also considered if a fixation met one of three types of descriptions: (1) linear drift (LD), (2) a "bumpy" fixation ("BF"), or a noisy fixation ("N"). For examples of fixations with linear drift see the first two fixations in Fig 14. A "bumpy" fixation was a fixation with low-frequency noise. There are ten examples shown in Fig 20. A "noisy" fixation was a

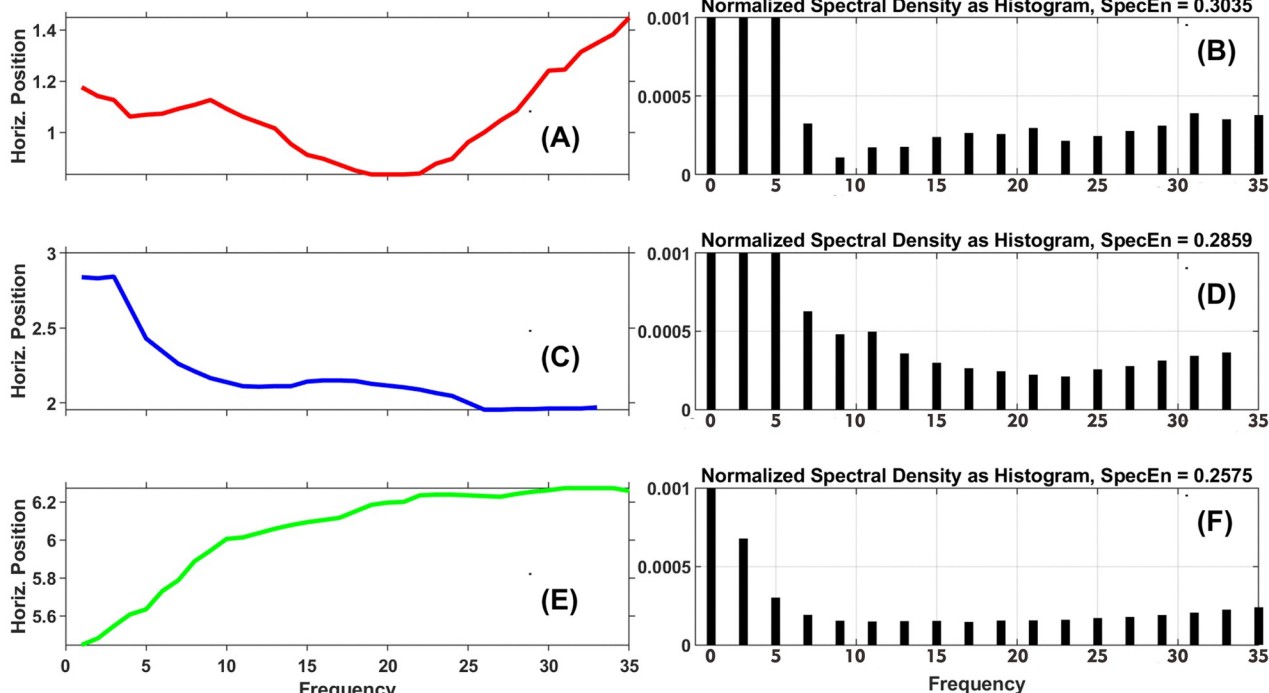

**Fig 10. Explanatory plot for the Spectral Entropy.** The frequency plots have been scaled from 0 to 0.001 to better visualize the higher frequency estimates. In Fig 10A, we present a relatively chaotic fixation, and its corresponding frequency plot is shown in Fig 10B. The chart of normalized spectral density by frequency exhibits multiple levels, resulting in a high value of SpecEn. In Fig 10C, we present a fixation that is much less chaotic, leading to a narrower range of normalized density in the higher frequencies (refer to Fig 10D). In Fig 10E, we showcase a relatively smooth fixation, which is associated with very little variability in normalized density estimates in the upper frequencies (see Fig 10F).

fixation with high-frequency noise. These fixations were most common in the OK Lab dataset (e.g., see Fig 21). This high-frequency noise may be ocular microtremor [85].

After each author rated the fixations, they met to reach a consensus on the rating of each fixation. The consensus ratings are presented in Figs 11 to 24.

In the GazeCom dataset, there are 14 fixations with high FuzzEn, which include either a portion or the entirety of a saccade (see Fig 18). On the other hand, Low FuzzEn is associated with nine fixations that start too late (see Fig 19).

The table with all ratings of each fixation is available (see Data Availability Statement at end). The next step in the High-Low analysis was to identify ratings that occurred eight or more times on a single page. If a page had eight or more occurrences of the same rating, we considered that rating to be characteristic of the level of entropy (high or low) for a particular dataset. This is presented as Table 7.

## Results

Table 7 summarizes the main results of this study. We found that high and low entropy values indicate different things for different datasets. In Table 7, we only include entropies where either high entropy, or low entropy, or both have some diagnostic value. For the GazeCom dataset, all six entropies were associated with some characterization of fixations. In the Lund dataset, only GridEn had any diagnostic value. In the OK Lab dataset, GridEn, SampEn, SpecEn, and FuzzEn had diagnostic value.

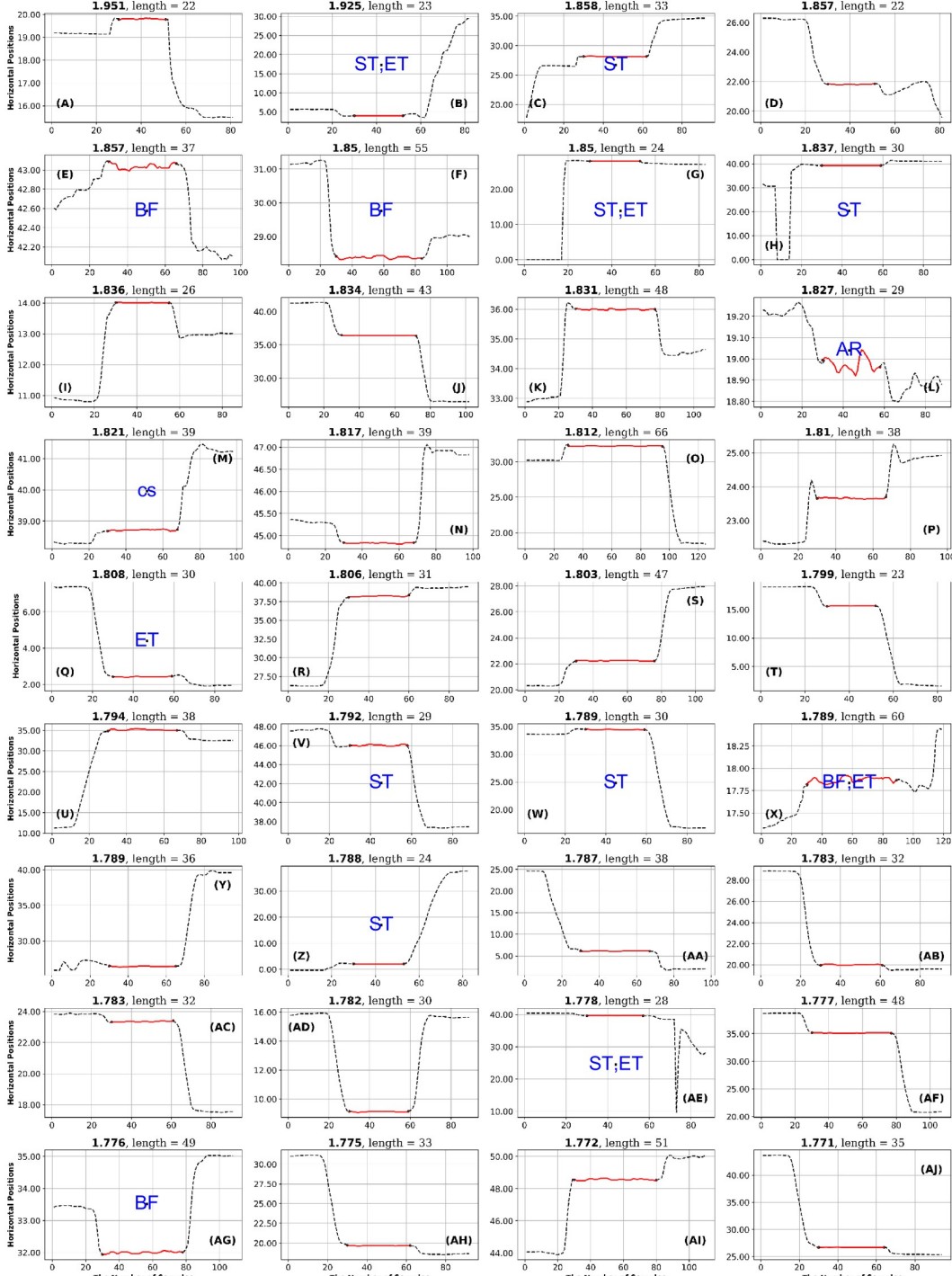

**Fig 11. GazeCom dataset: 36 horizontal fixations with the highest GridEn values.** The fixations are plotted in red. There are 30 signal samples before and after each fixation. The entropy value is displayed above each fixation chart. The classification of each fixation is shown in blue in the middle of each fixation chart. The fixation at the top left has the highest entropy value. Note the common occurrence of fixations that start too late on this page.

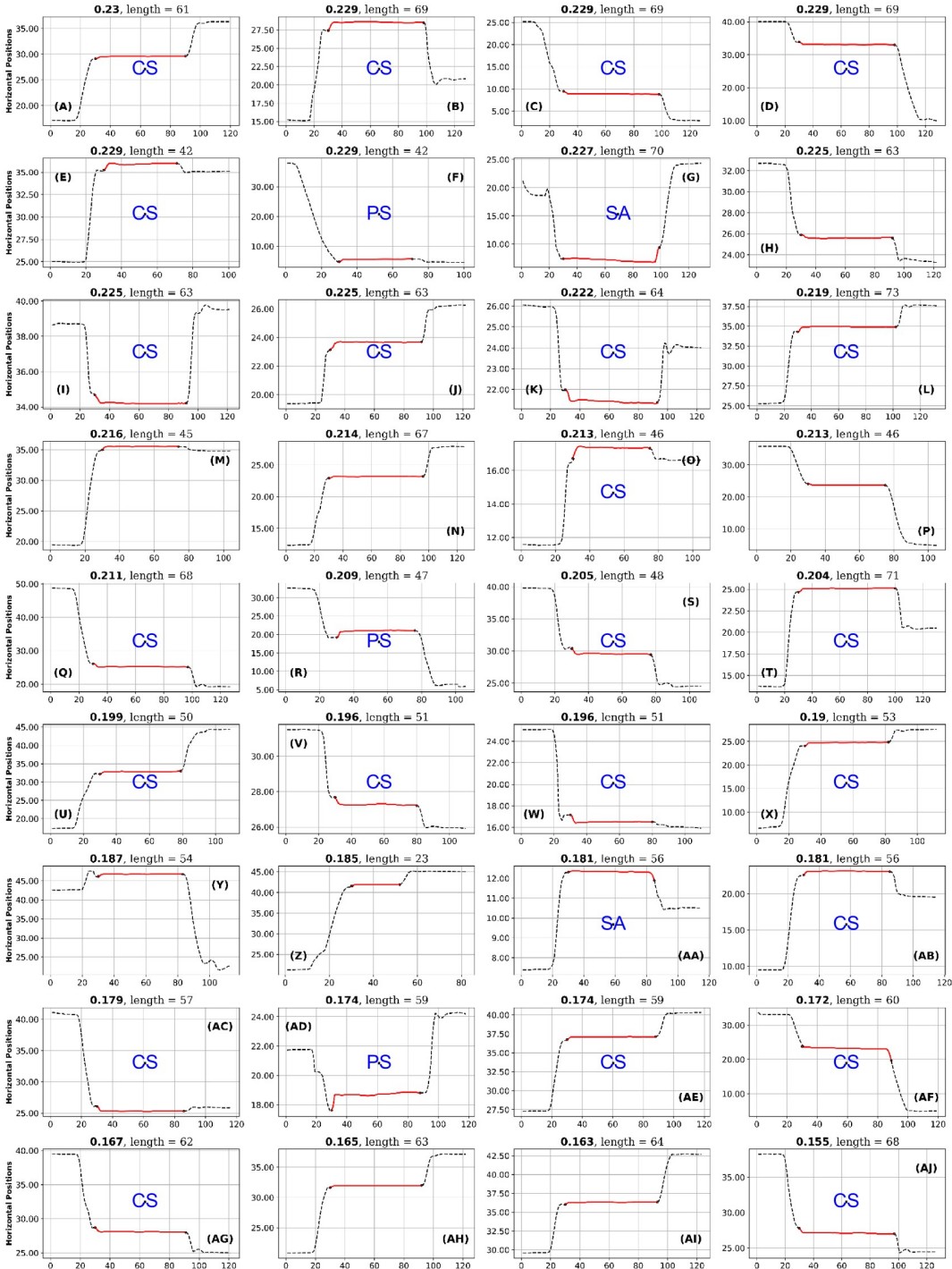

**Fig 12. GazeCom dataset: 36 fixations with the lowest GridEn values.** See the caption of Fig 11 for more details. The fixation with the lowest entropy value is shown in the bottom right. Note that 23 fixations start with a corrective saccade (CS) on this page.

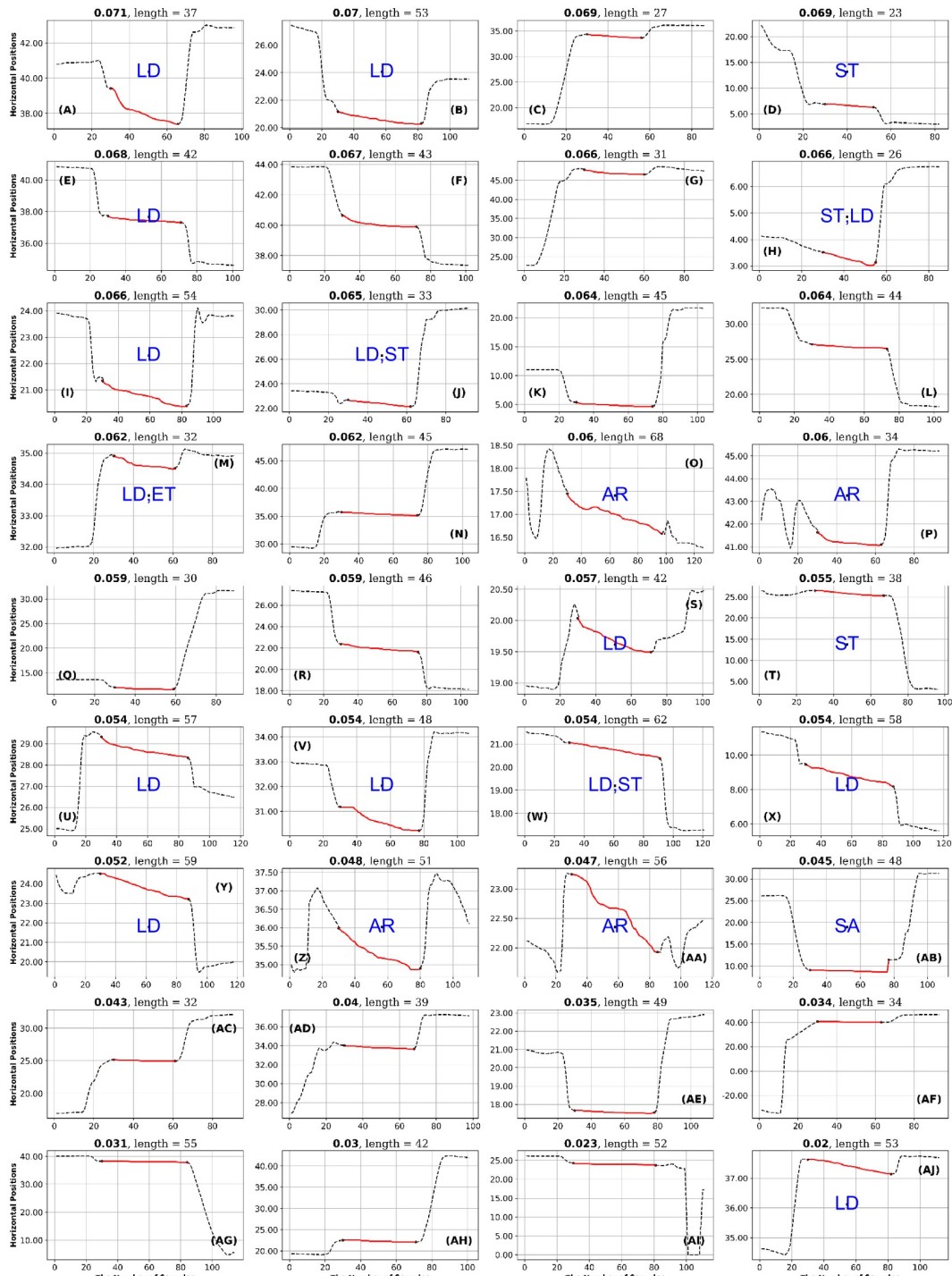

**Fig 13. GazeCom dataset: 36 fixations with the lowest PhasEn values.** See the caption of Fig 11 for more details. Note that 14 fixations exhibit linear drift (LD) on this page.

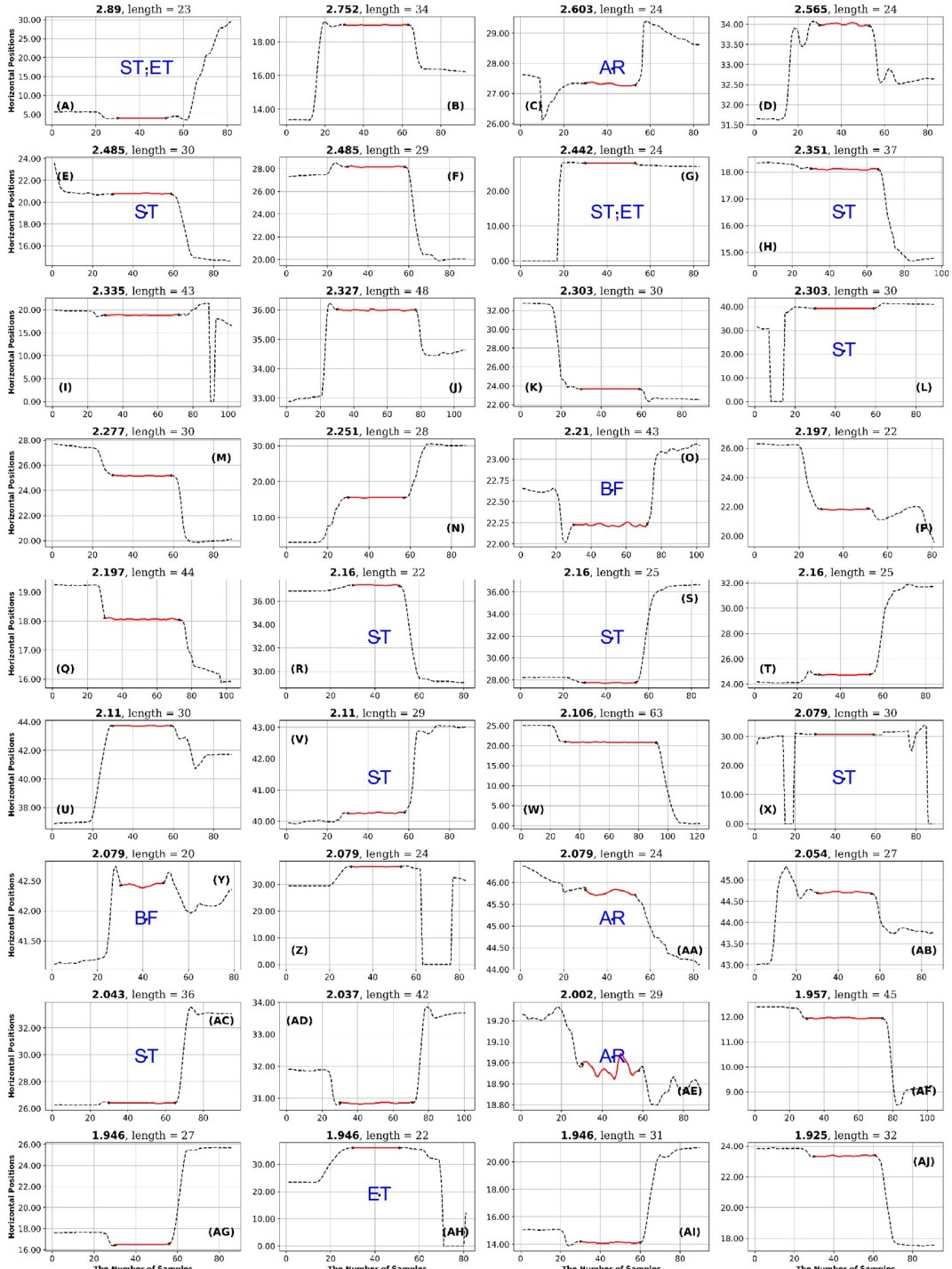

**Fig 14. GazeCom dataset: 36 fixations with high SampEn values.** See the caption of Fig 11 for more details.

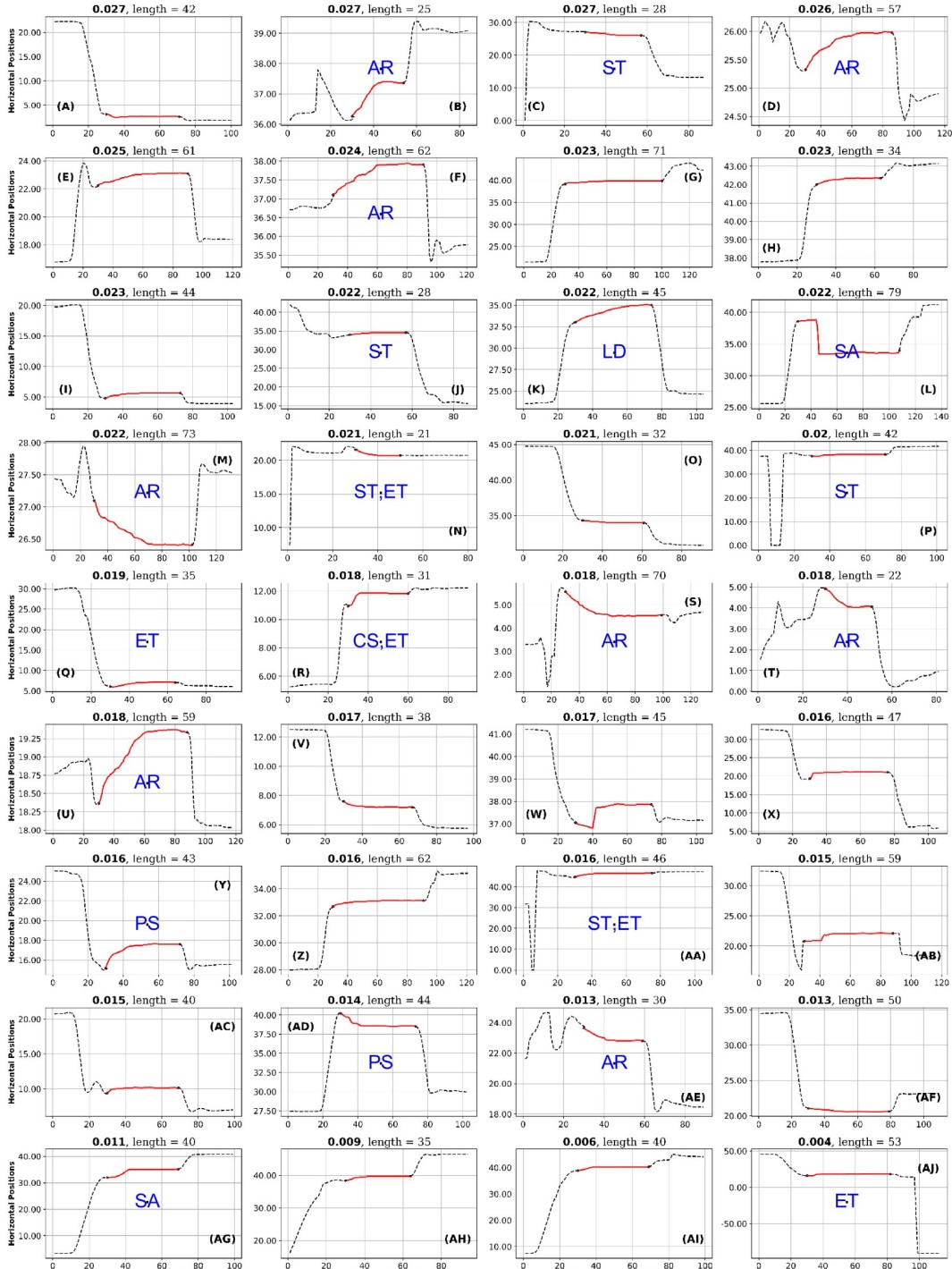

**Fig 15. GazeCom dataset: 36 fixations with low SampEn values.** See the caption of Fig 11 for more details.

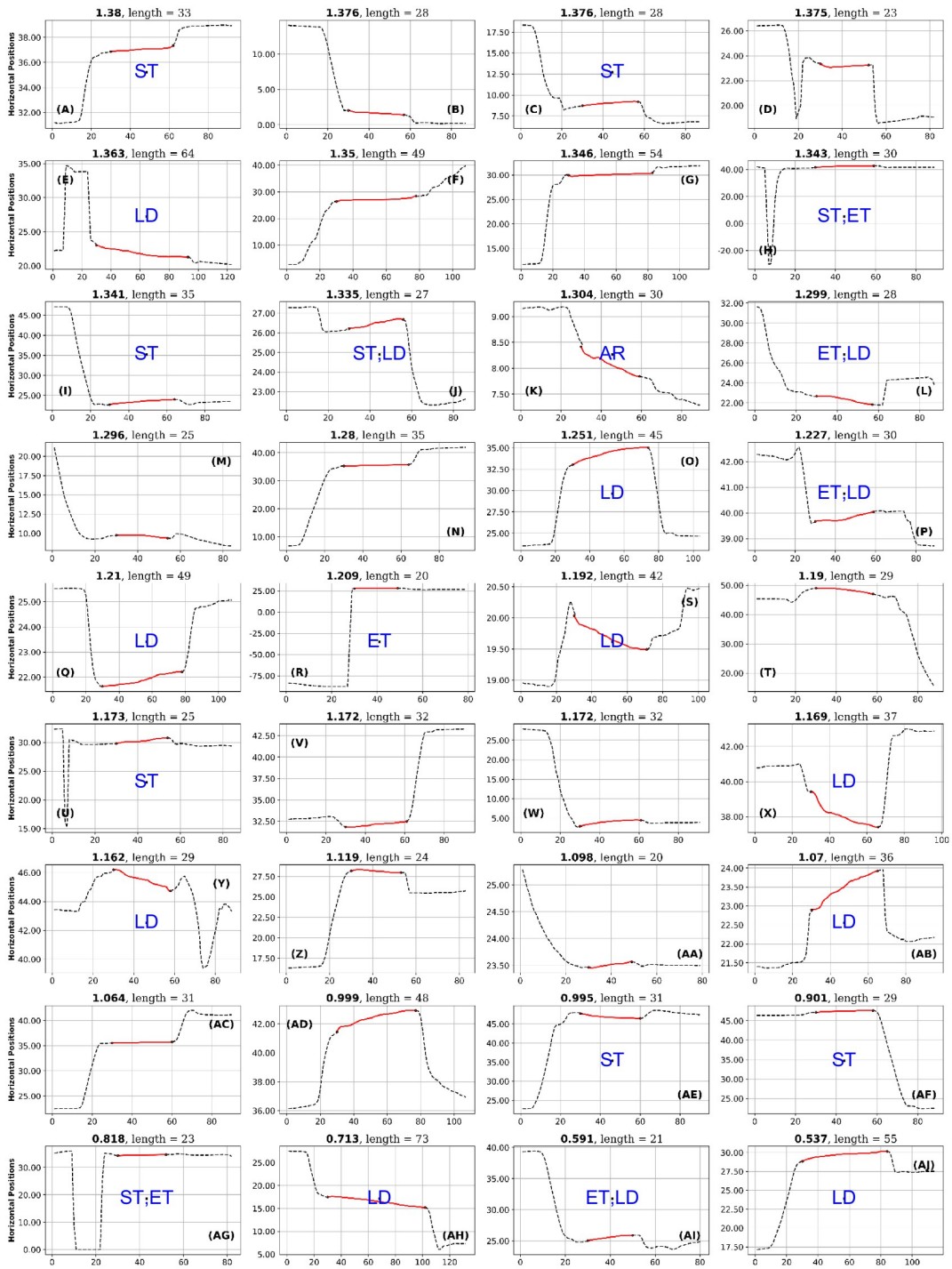

**Fig 16. GazeCom dataset: 36 fixations with low IncrEn values.** See the caption of Fig 11 for more details.

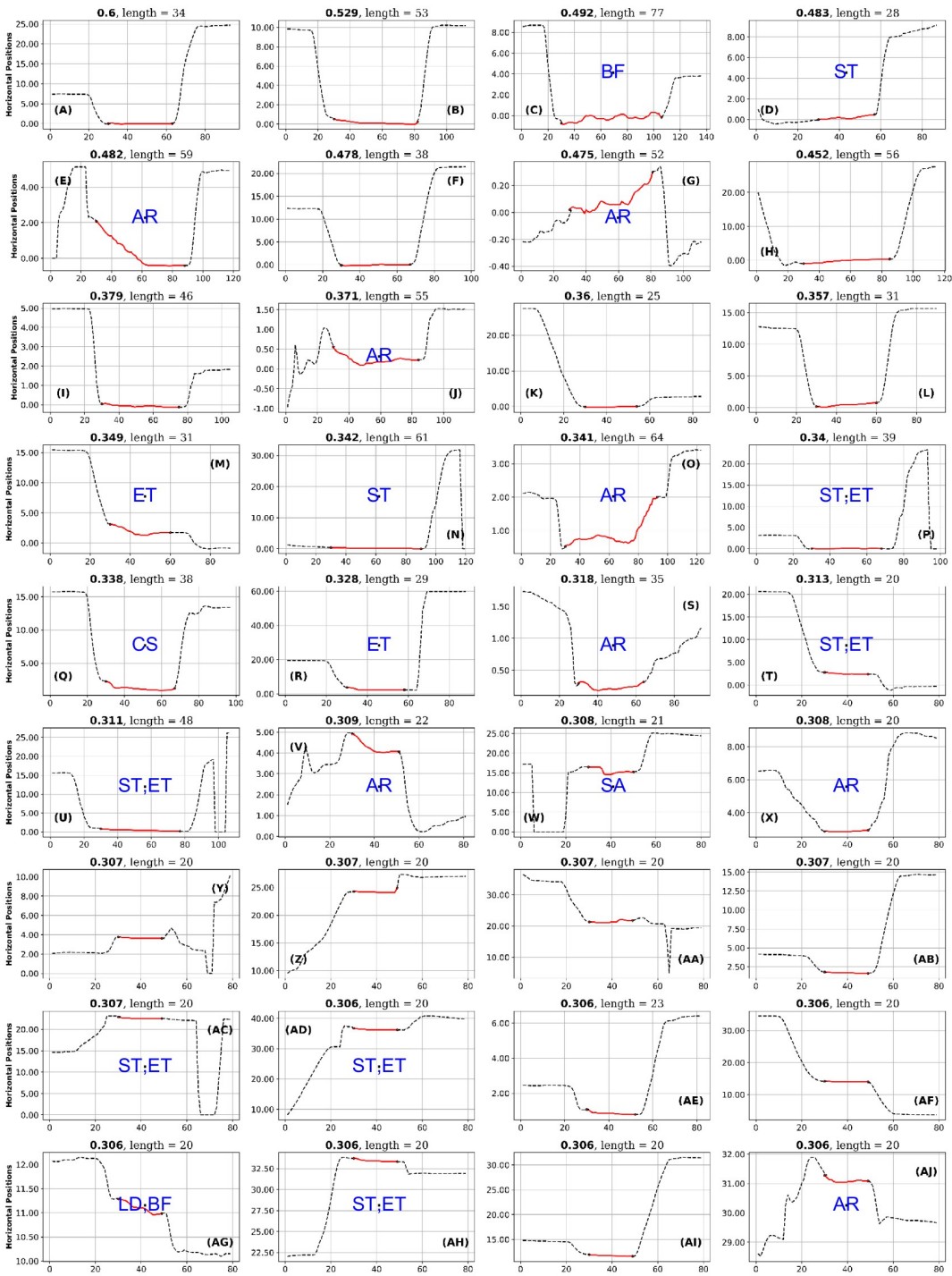

**Fig 17. GazeCom dataset: 36 fixations with high SpecEn values.** See the caption of Fig 11 for more details.

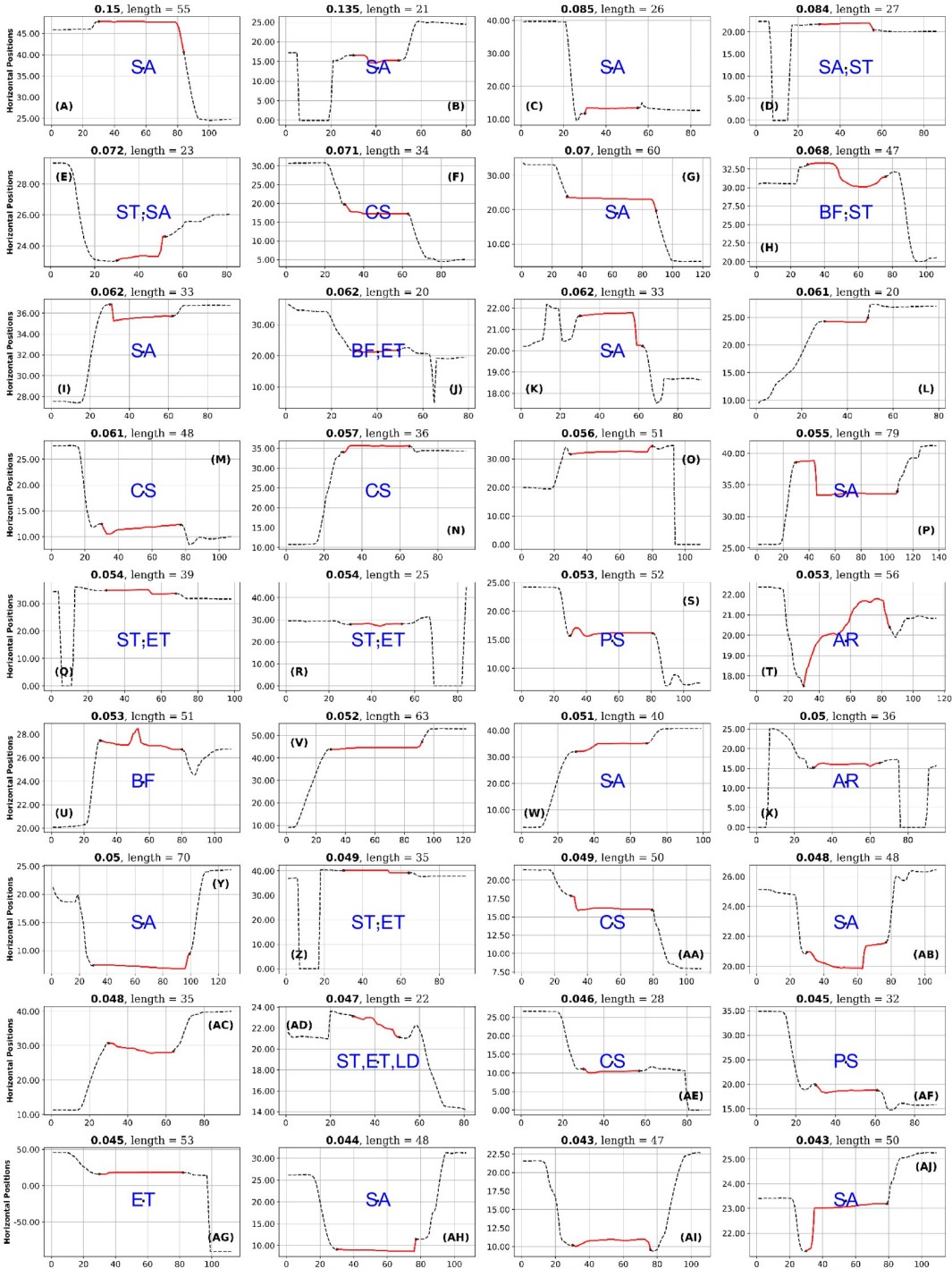

**Fig 18. GazeCom dataset: 36 fixations with high FuzzEn values.** See the caption of Fig 11 for more details.

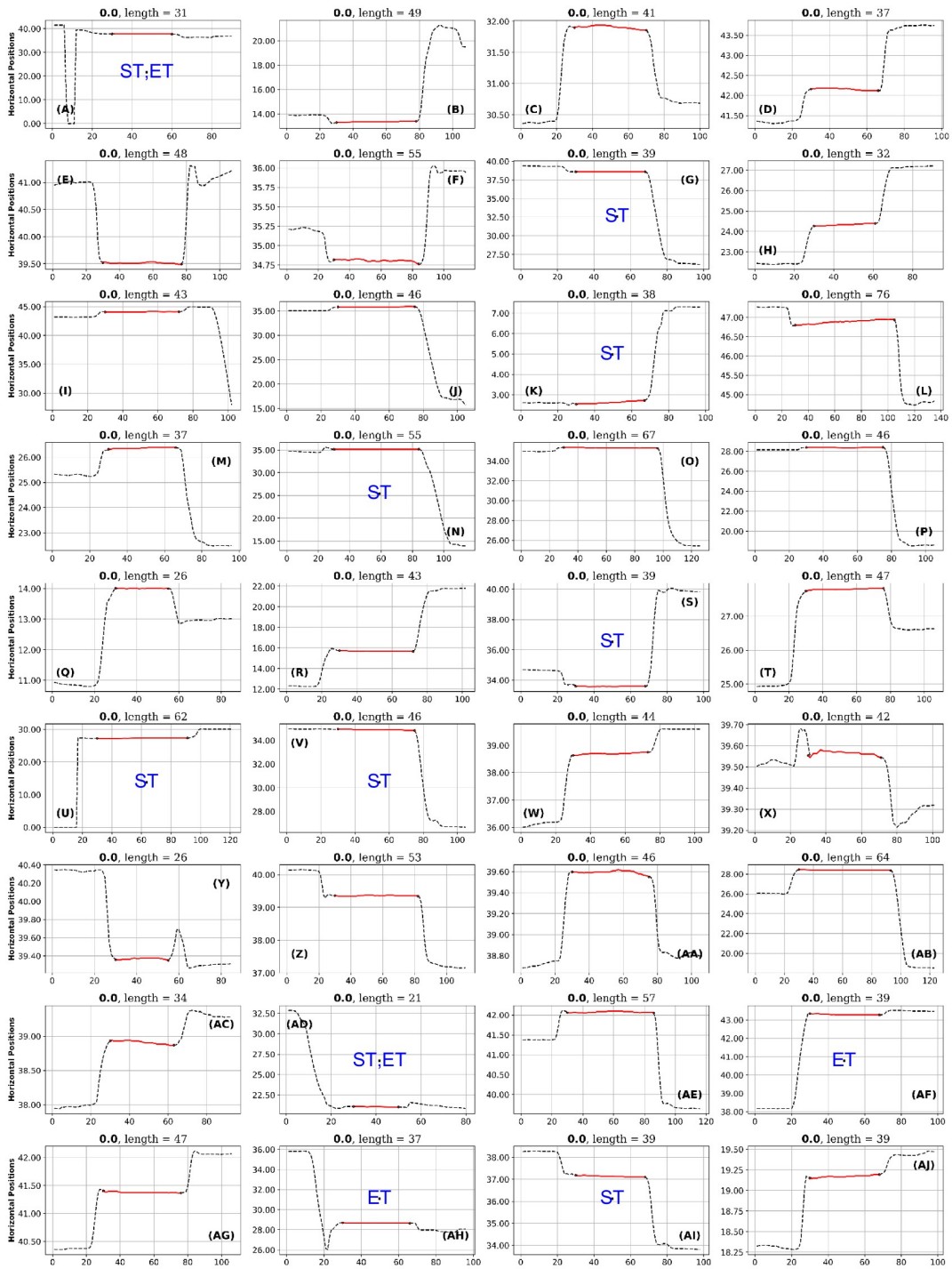

**Fig 19. GazeCom dataset: 36 fixations with low FuzzEn values.** See the caption of Fig 11 for more details.

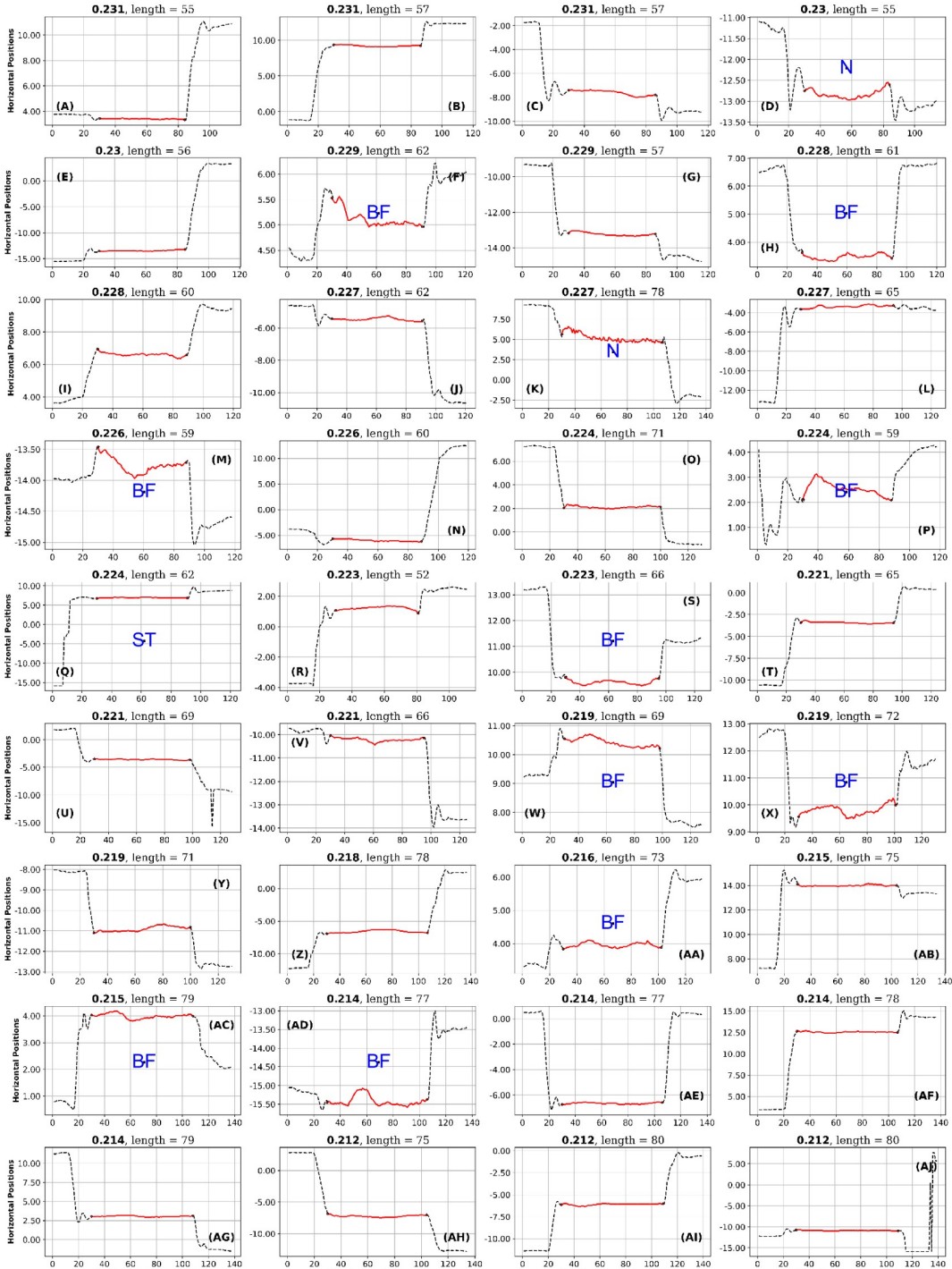

**Fig 20. Lund dataset: 36 fixations with low SpecEn values.** See the caption of Fig 11 for more details.

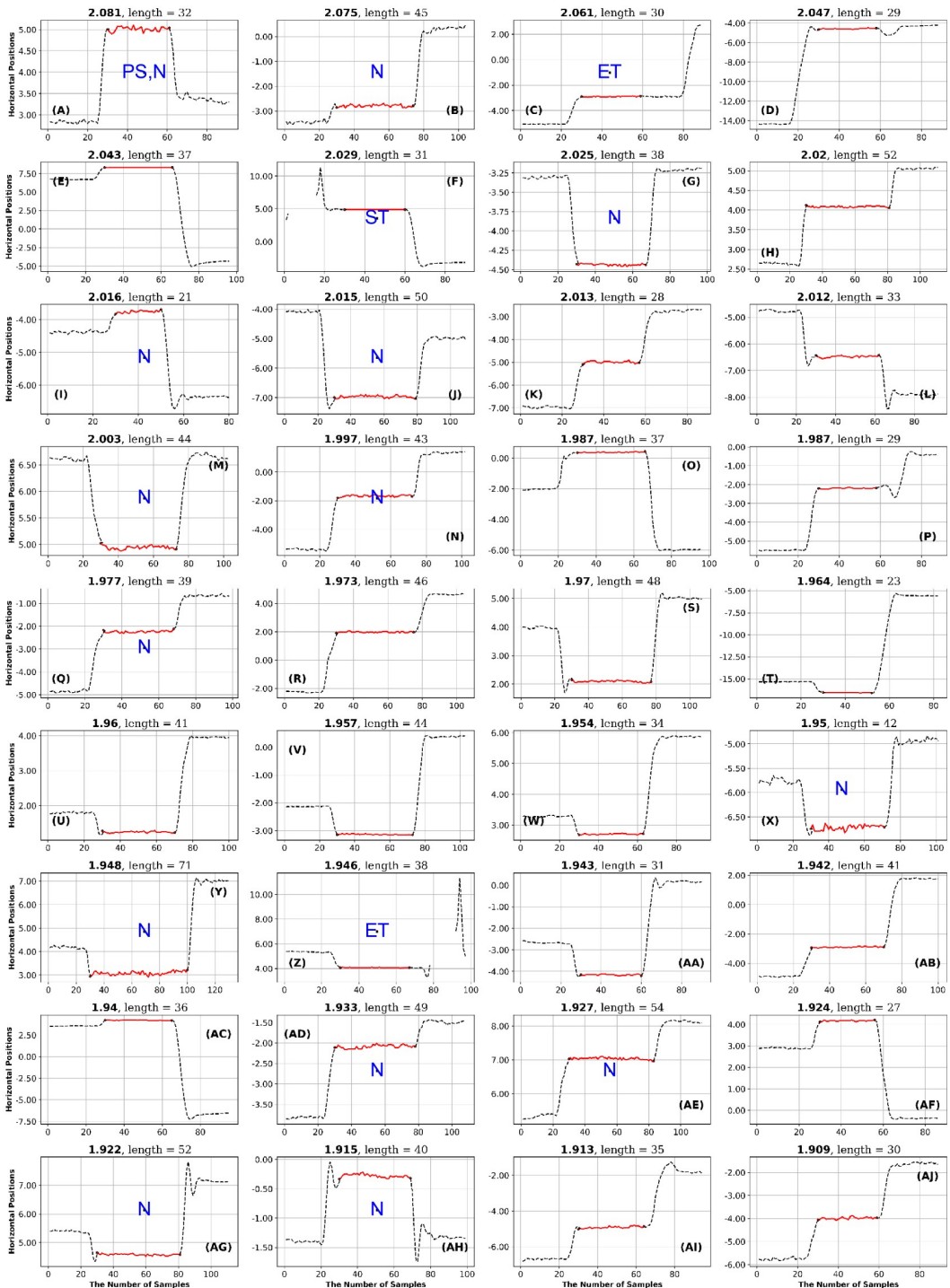

**Fig 21. OK lab dataset: 36 fixations with high GridEn values.** See the caption of Fig 11 for more details.

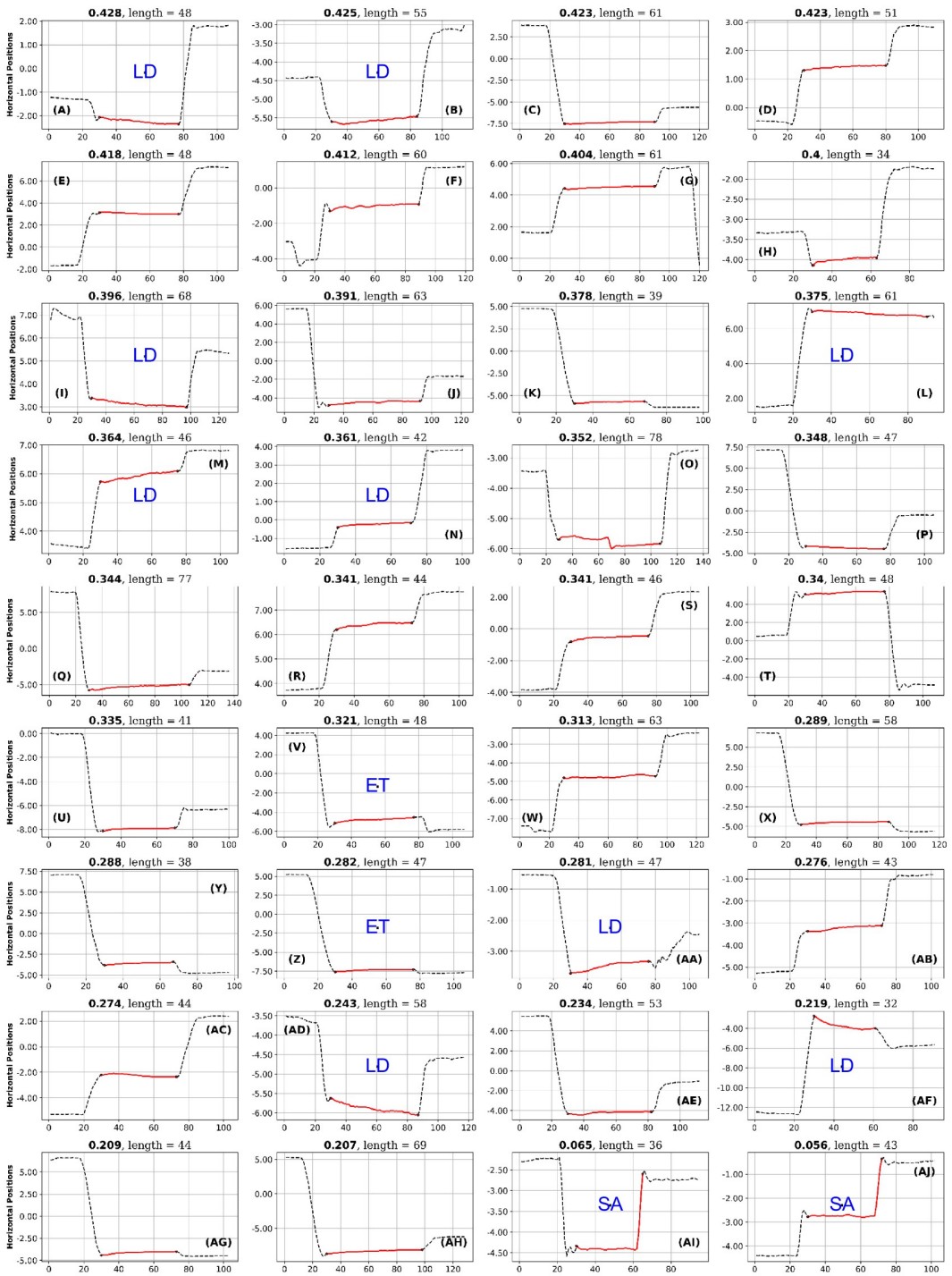

**Fig 22. OK lab dataset: 36 fixations with low SampEn values.** See the caption of Fig 11 for more details.

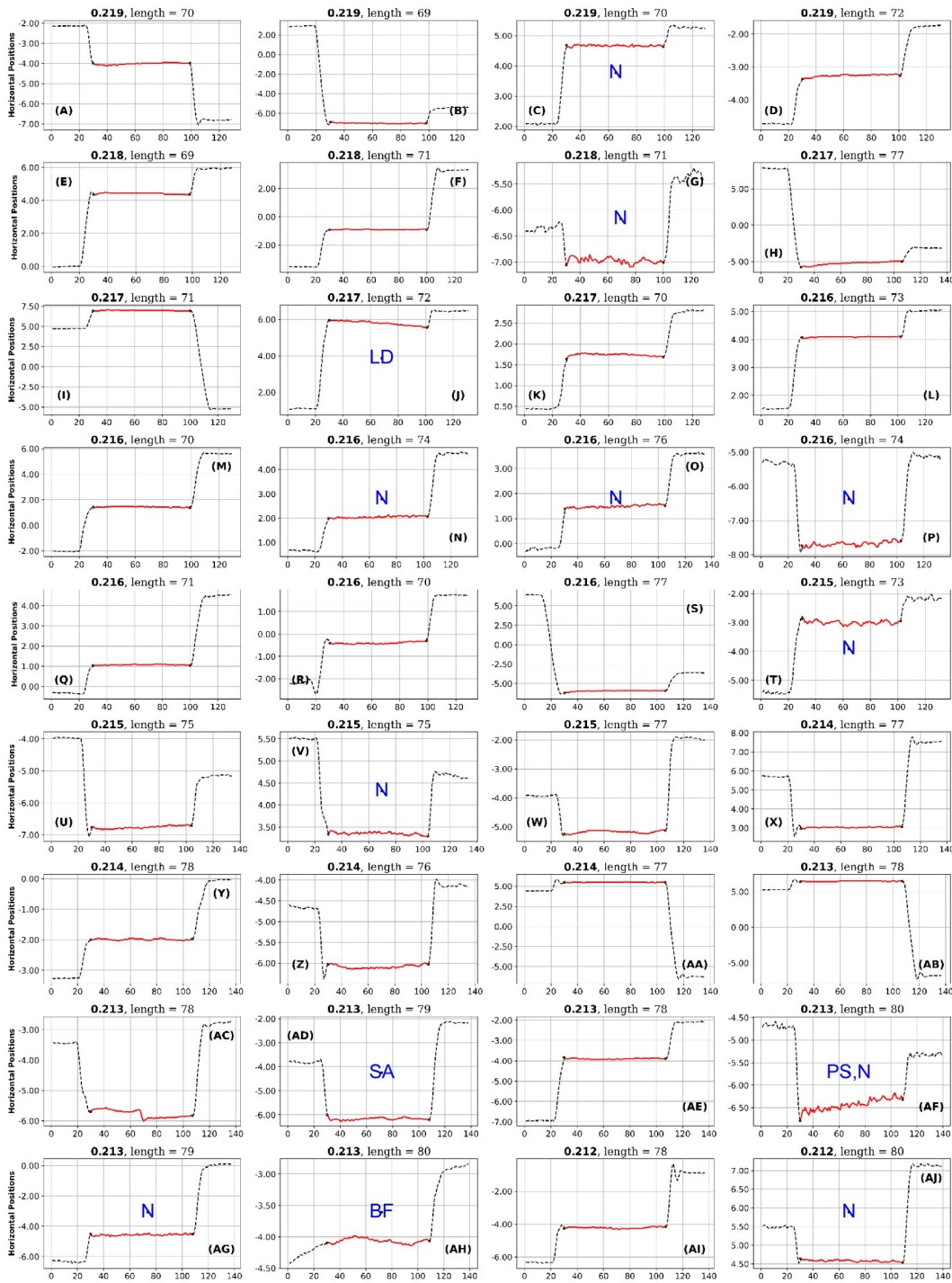

**Fig 23. OK lab dataset: 36 fixations with low SpecEn values.** See the caption of Fig 11 for more details.

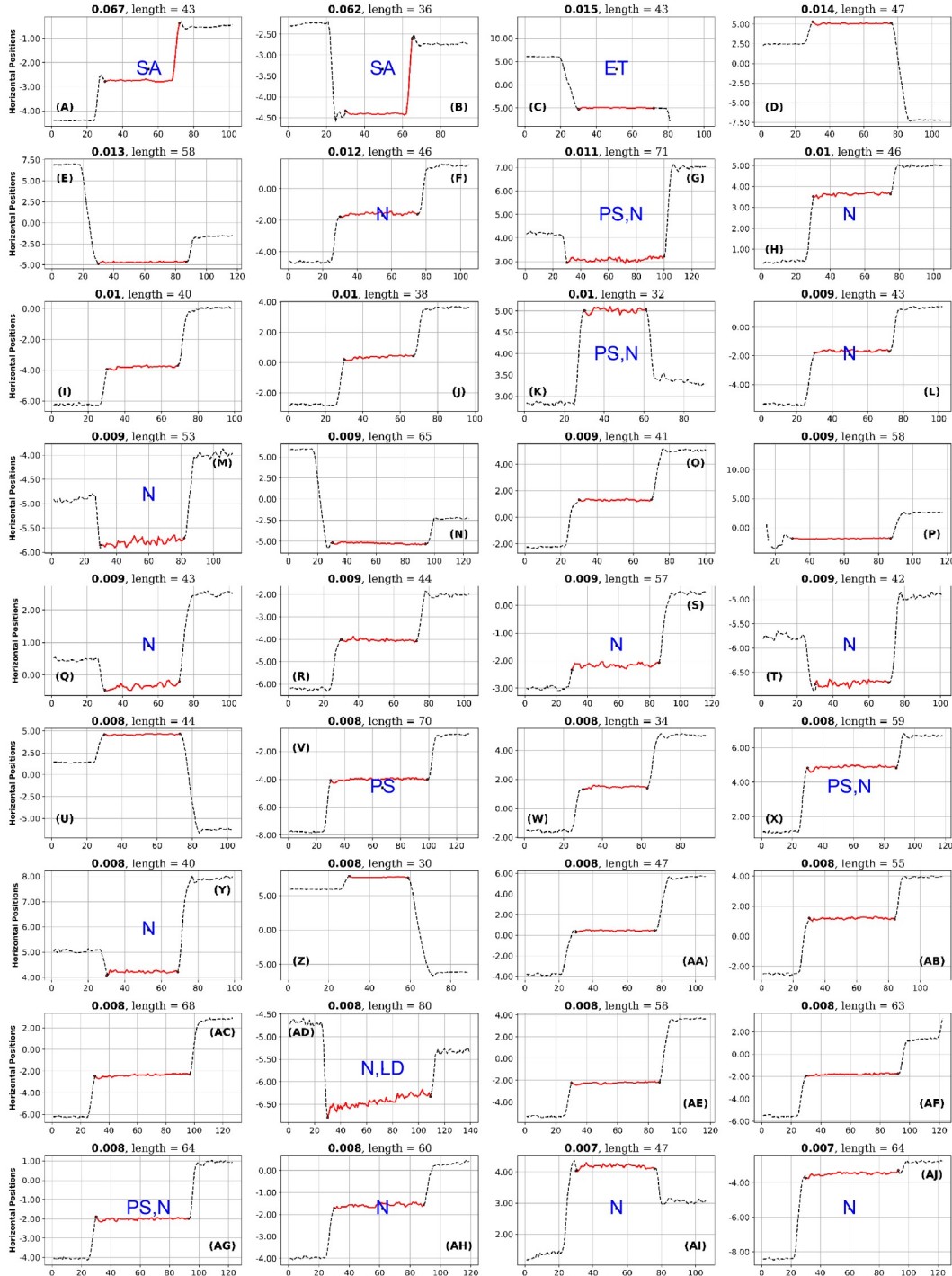

**Fig 24. OK lab dataset: 36 fixations with high FuzzEn values.** See the caption of Fig 11 for more details.

**Table 6. List of codes used to classify the fixations.**

| Misclassification Codes | Meaning |
|---|---|
| ST | Fixation starts too late. |
| ET | Fixation ends too early. |
| CS | Fixation starts with a corrective saccade. |
| PS | Fixation starts with either an entire PSO or part of a PSO. |
| AR | Fixation is associated with signal artifact. |
| SA | Fixation has a complete or partial saccade. |

| Descriptive Codes | Meaning |
|---|---|
| LD | Fixation with a linear drift. |
| BF | Bumpy fixation |
| N | High-frequency noise in fixation |

## GazeCom dataset

A global examination of Table 7 indicates that all misclassifications of fixations occur in the GazeCom dataset. A common problem in the GazeCom dataset was that fixations started too late (code ST). Five of the twelve pages had eight or more fixations that started too late.

In the GazeCom dataset, high GridEn was associated with saccades that start too early (Table 7 and Fig 11). On the other hand, the low GridEn page had 23 fixations out of a total of 36 that started with corrective saccades.

In the GazeCom dataset, high PhaseEn is not associated with any particular misclassification or fixation description. However, a low PhaseEn is associated with a set of 14 fixations that display a linear drift (see Fig 13).

In the GazeCom dataset, on the high SampEn fixation page (see Fig 14), 10 fixations start too late. Eight of the fixations on the low SampEn page are contaminated with artifact (see Fig 15).

**Table 7. Results of high low analysis.**

| Dataset | Entropy Type* | ST | ET | CS | PS | AR | SA | LD | BF | N |
|---|---|---|---|---|---|---|---|---|---|---|
| GazeCom | HE GridEn | 8 | | | | | | | | |
| GazeCom | LE GridEn | | | 23 | | | | | | |
| GazeCom | LE PhasEn | | | | | | | 14 | | |
| GazeCom | HE SampEn | 10 | | | | | | | | |
| GazeCom | LE SampEn | | | | | 8 | | | | |
| GazeCom | LE IncrEn | 9 | | | | | | 13 | | |
| GazeCom | HE SpecEn | 8 | 8 | | | 8 | | | | |
| GazeCom | HE FuzzEn | | | | | | 14 | | | |
| GazeCom | LE FuzzEn | 9 | | | | | | | | |
| Lund | LE SpecEn | | | | | | | | 10 | |
| OK_Lab | HE GridEn | | | | | | | | | 14 |
| OK Lab | LE SampEn | | | | | | | 9 | | |
| OK Lab | LE SpecEn | | | | | | | | | 10 |
| OK Lab | HE FuzzEn | | | | | | | | | 16 |

*—HE: High Entropy, LE: Low Entropy

High IncrEn in the GazeCom dataset is not associated with any fixation issue. However, low IncrEn is associated with fixations that start too late and exhibit significant linear drift (see Fig 16).

High SpecEn in the GazeCom dataset is associated with fixations that start too late, fixations that end too early, and fixations affected by artifacts (see Fig 17). In this dataset, low SpecEn is not associated with any unusual fixations.

In the GazeCom dataset, there are 14 fixations with high FuzzEn, which include either a portion or the entirety of a saccade (see Fig 18). On the other hand, Low FuzzEn is associated with nine fixations that start too late (see Fig 19).

### Lund dataset

As noted above, only a single entropy, SpecEn, had anydiagnostic value for the Lund dataset. In this case, Low SpecEn produced 10 bumpy fixations (see Fig 20). High SpecEn was not diagnostic.

### OK Lab dataset

As noted above, four of the six final entropies had some diagnostic value for the OK Lab Dataset. High levels of GridEn and FuzzEn indicated fixations with high-frequency noise (see Table 7, Figs 21 and 24). Low SpecEn also produces 10 noisy fixation examples (Fig 23). Low SampEn is associated with nine fixations with linear drift (Fig 22).

## Discussion

The overall goal of this report was to evaluate what kind of information entropy metrics provide about eye movement fixations. We have evaluated this in 3 datasets: (1) GazeCom, (2) Lund, and (3) the OK Lab. The main result is that entropy metrics detect different fixation characteristics in each dataset. There are no conclusions regarding entropy measures that apply across datasets.

We took several steps to ensure that our evaluation of each dataset was based on comparable underlying data. The GazeCom dataset was sampled at 250 Hz, the Lund dataset was sampled at 500 Hz and the OK Lab dataset was sampled at 1000 Hz. We downsampled both the Lund dataset and the OK Lab dataset to 250 Hz to make them more comparable to the GazeCom dataset. We also equalized fixation durations across the datasets. But there are other characteristics of the three datasets that could not be equalized. The GazeCom subjects viewed video clips, the Lund subjects viewed static images and the OK Lab subjects were reading poetry. Also, the number of fixations differed widely across datasets. After duration equalization, the GazeCom dataset had 20,613 fixations, the Lund dataset had 145 fixations and the OK Lab dataset had 874 fixations. The 72 fixations we based our high-low analysis on corresponded to 0.35% of the GazeCom dataset, 49.7% of the Lund dataset, and 8.2% of the OK Lab dataset. We could have equalized these percentages by drawing random sub-samples from the GazeCom and OK Lab datasets. In one sense such an analysis might have been more fair. However, we reasoned that such an equalization would produce a less useful analysis of the information provided by entropy metrics.

The entropy metrics of fixations in the GazeCom dataset found a number of fixations that the present authors consider to be misclassified. The types of misclassifications we found in this dataset include fixations saccades that start too late, fixations that include an initial corrective saccade, fixations that include, or are influenced by signal artifact, fixations that include all or part of a saccade, and fixations that end too early. When interpreting this finding it is important to keep in mind that we are evaluating 0.35% of the fixations available in this

dataset. On this basis, we cannot make any statement about the rates of misclassification in the GazeCom dataset as a whole. However, it is useful to review how classifications of eye movements were made in this dataset. Apparently, a large part of the classification work was performed by two novice annotators who were paid undergraduate students. Although eye movement experts were available and involved at several stages, the Lund dataset was classified by a single individual (Richard Andersson) who is an internationally known expert in eye movement classification. The OK Lab dataset was classified by an individual who had undergone very extensive training as outlined in [55]. Our entropy analysis of fixations in the Gaze-Com dataset might be useful for future users of these recordings. High and low entropy metrics might well be useful in an effort to improve the classification of fixation in the Gaze-Com dataset.

In addition, for GazeCom, two entropy measures (low PhasEn and low IncrEn) detected fixations that have marked linear drift. The movie-clip stimuli used for the GazeCom dataset did induce some smooth pursuit. Perhaps these entropy measures might assist in determining if some of the "fixations" detected with marked linear drift are actually periods of smooth pursuit.

For the Lund dataset, with far fewer fixations, only one entropy type, SpecEn, had any diagnostic value. Low SpecEn tended to find bumpy fixations, which have low-frequency oscillations. As far as we are aware, we are the first to distinguish such fixations, and we are not aware of prior research on the physiological basis or interpretation of such fixations. Although occasional bumpy fixations were noted in both the GazeCom and the OK Lab datasets, in no case were there eight or more bumpy fixations detected by any form of entropy.

In the case of the OK Lab dataset, three entropy types (high GridEn, low SpecEn, and high FuzzEn) detected "noisy" fixations, which are characterized by high-frequency oscillations. Although McCamy et al [85] state that ocular microtremor of fixation cannot be detected with the video-oculography (VOG) method, we think that this might not always be the case. The most likely interpretation of these noisy fixations is that they are examples of ocular microtremors that are detectable in some fixations from some individuals. The OK Lab data were collected on an EyeLink 1000, which is known to be a very accurate device with high precision [86].

Low SampEn detected fixations with linear drift in the OK Lab dataset. It is interesting to note that this is not the entropy that detected fixations with linear drift in the GazeCom dataset (low PhaseEn, low IncrEn). The recorded subjects in the OK Lab dataset were reading a poem. Such a stimulus does not induce smooth pursuit. These examples of linear drift in fixations might just be part of the normal variation in fixation stability. This also applies to the fixations with linear drift in the GazeCom dataset, although the latter did use stimuli that induced some periods of smooth pursuit.

All of these observations apply only to fixations that are longer than 40 msec. For GazeCom, we rejected 95 fixations or 0.26% of all fixations because they were too short. There were no fixations in either the Lund or the OK Lab datasets that were less than 40 msec. It is worth noting that Nyström and Holmqvist [87] set 40 msec as the minimum duration of fixation in their eye movement classification algorithm. Therefore, it appears that this limitation had a very limited impact on our analysis.

In summary, we applied six entropy metrics to the horizontal fixation signals from three different datasets (GazeCom, Lund, and OK Lab). The character of fixations identified by the entropy measures depended on the specific dataset. Therefore, it is not possible to draw a general conclusion about the significance of high or low entropy across these three datasets. In the GazeCom dataset, entropy measures detected certain misclassified fixations. However, given that we are only looking at a small portion of this dataset, we cannot make generalizations

across the whole set of fixations. For the Lund dataset, low SpecEn detected fixations with low-frequency noise ("bumpy fixations"). The oculomotor basis of such fixations is not clear at this time. For the OK Lab dataset, it appears that high GridEn, low SpecEn, and high FuzzEn detected fixations with high-frequency noise. This high-frequency noise is likely to be an ocular microtremor that occurs in some subset of subjects and fixations.

## Author Contributions

**Conceptualization:** Kateryna Melnyk.

**Formal analysis:** Kateryna Melnyk, Lee Friedman.

**Funding acquisition:** Oleg V. Komogortsev.

**Investigation:** Kateryna Melnyk, Lee Friedman.

**Methodology:** Kateryna Melnyk, Lee Friedman.

**Project administration:** Oleg V. Komogortsev.

**Resources:** Oleg V. Komogortsev.

**Software:** Kateryna Melnyk, Lee Friedman.

**Supervision:** Lee Friedman, Oleg V. Komogortsev.

**Visualization:** Kateryna Melnyk, Lee Friedman.

**Writing – original draft:** Kateryna Melnyk.

**Writing – review & editing:** Lee Friedman, Oleg V. Komogortsev.

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
