## [Decision Letter · Decision Letter 0]

10 Apr 2023

PONE-D-22-33922What can Entropy Metrics tell us about types of fixations during video viewing?PLOS ONE

Dear Dr. Melnyk,

Thank you for submitting your manuscript to PLOS ONE. I apologise for the prolonged review period. After careful consideration, we feel that it has merit but does not fully meet PLOS ONE’s publication criteria as it currently stands. Therefore, we invite you to submit a revised version of the manuscript that addresses the points raised during the review process.

We look forward to receiving your revised manuscript.

Kind regards,

Patricia Wollstadt, Ph.D.

Academic Editor

PLOS ONE

Journal Requirements:

The study was funded by 3 grants to Dr. Komogortsev: (1) National Science 608

Foundation, CNS-1250718 and CNS-1714623, www.NSF.gov; (2) National Institute of 609

Standards and Technology, 60NANB15D325, www.NIST.gov; (3) National Institute of 610

Standards and Technology, 60NANB16D293. 

However, funding information should not appear in the Acknowledgments section or other areas of your manuscript. We will only publish funding information present in the Funding Statement section of the online submission form. 

(1) O.V.K., National Science 608Foundation, CNS-1250718 and CNS-1714623, www.NSF.gov; 

(2) O.V.K., National Institute of 609Standards and Technology, 60NANB15D325, www.NIST.gov; 

(3) O.V.K., National Institute of 610Standards and Technology, 60NANB16D293, www.NIST.gov; 

he funders had no role in study design, data collection and analysis, decision to publish, or preparation of the manuscript.

Reviewers' comments:

Reviewer's Responses to Questions

**Comments to the Author**

1. Is the manuscript technically sound, and do the data support the conclusions?

Reviewer #1: Yes

2. Has the statistical analysis been performed appropriately and rigorously? 

Reviewer #1: Yes

3. Have the authors made all data underlying the findings in their manuscript fully available?

Reviewer #1: Yes

4. Is the manuscript presented in an intelligible fashion and written in standard English?

Reviewer #1: No

5. Review Comments to the Author

Reviewer #1: General comments

This manuscript provides a valuable overview of different entropy measures for fixation analysis during video viewing. Using the GazeCom database that includes gaze data for 18 video clips à 20 s each, the authors apply established entropy measures and evaluate their potential to characterize fixation trajectories. This approach and the summary of findings appears very useful for advancing gaze analyses in future studies focusing on viewing patterns in natural scenes. The overall outline of the manuscript is clear and technical information is mostly presented coherently. However, I have some suggestions how to improve the rationale structure and carve out the main insights more concisely. In addition, some general statements throughout the manuscript could be tuned down or phrased more cautiously.

Specific comments

Abstract: the abstract could be shortened significantly; in particular, the second half gives detailed information, e.g., on specific entropy features, that are hard to follow and need the introduction provided only later on. Thus, I suggest to focus just on the technical outline of the study and to summarize the key conclusions on a more abstract level. The current concluding sentences seem quite fuzzy (Does the work really aims just at *this*, i.e., GazeCom, dataset? Is a focus on the differentiation between fixations and smooth pursuit really justified given the later discussion in the manuscript? Could the value for classification system be described more specifically?)

l.2ff the first sentence seems quite self-confident – I just encourage to rethink whether this strong claim should really be kept up

l.15ff maybe readers that do not have already advanced knowledge of chaos theory and entropy measures could profit from an overview table that characterizes cores measures; here and in the following sections a variety of technical terms is taken up that seems hard to follow without at least a rough classification.

l.130ff this paragraph should be revised in order to give a concise introduction the following Methods section. Consider moving the information on the choice of entropy metrics and high-low analyses to the Methods section where the details are provided; I assume that here many readers might be just puzzled because it is not possible to understand the choice of methods. The general aim of the study (last sentence of the paragraph) could be phrased stronger. Also note that there are inconsistencies between statements in this paragraph and later on in the Methods section (see reasoning for choosing the GazeCom database).

Methods:

I would appreciate a careful revision of the Methods section, including the Appendix material (additional note: the parallel offer of Appendix and Supplementals might be confusing). In the current version there is a plethora of figure in which the audience might get lost. Ideally it would be possible to re-evaluate which figures a really critical for getting the key findings. I would prefer less figures explained and documented in more detail; e.g., I think the figures in the complex facets of the Appendix’ might not be self-explanatory.

Table 2: Here you give the six selected measures, but the reasoning for the choice is only given in the Results section; the logical order should be improved.

l.242ff the rationale for providing an elaborated description of some entropy measures does not get clear; *base entropy measures* seems to have only one subsection; the selection of different features is not motivated here (see first comment Methods)

l.334ff is it possible to explain the choice of *36* fixations? The statement that the manuscript cannot be understood without the Appendix’ figures seems problematic – in my understanding an Appendix should provide additional information, not core information. I think it is necessary to prepare a comprehensible summary of relevant issues. Personal references to co-authors and their appraisal appear unnecessary – the readers will form their opinion about the validity of the applied procedure.

Results:

The presentation of results seems not well-structured.

ll.356ff the reader would appreciate more than just *some idea*. The organization of the results section is not convincingly laid out and is also not reflected in the later paragraphs.

ll.377ff *that looked most promising* - sorry, that statement needs more substance

Table 6 it would be favorable if the order in which the measures are presented could be consistent in the Table and the text.

Discussion:

The overall discussion of the provided insights could be more focused. Given the very stringent methods that the presented results are based on, it appears discrepant that conclusions remain rather arbitrary in the discussion section (e.g., *we don’t really know*, *perhaps more detailed analysis of parameter choices….*, and so on). What are the main conclusions? What should the audience take along?

The general criticism concerning GazeCom appears fundamental – I am not sure whether the authors really intend to punch the dataset in total, but at the moment it sounds a little bit like this. It would be interesting to know whether the authors tried to get into contact with GazeCom developers and have discussed these issues.

6. PLOS authors have the option to publish the peer review history of their article (what does this mean?). If published, this will include your full peer review and any attached files.

Reviewer #1: No

---

## [Author Response · Author response to Decision Letter 0]

15 Aug 2023

See uploaded Review_Response.pdf file for the specific response to the reviewer comments.

---

## [Decision Letter · Decision Letter 1]

6 Sep 2023

What Can Entropy Metrics Tell Us About the Characteristics of Ocular Fixation Trajectories?

PONE-D-22-33922R1

Dear Dr. Melnyk,

We’re pleased to inform you that your manuscript has been judged scientifically suitable for publication and will be formally accepted for publication once it meets all outstanding technical requirements. Thank you for the careful revision of your manuscript.

Kind regards,

Patricia Wollstadt, Ph.D.

Academic Editor

PLOS ONE

Additional Editor Comments (optional):

Reviewers' comments:

Reviewer's Responses to Questions

**Comments to the Author**

1. If the authors have adequately addressed your comments raised in a previous round of review and you feel that this manuscript is now acceptable for publication, you may indicate that here to bypass the “Comments to the Author” section, enter your conflict of interest statement in the “Confidential to Editor” section, and submit your "Accept" recommendation.

Reviewer #1: All comments have been addressed

2. Is the manuscript technically sound, and do the data support the conclusions?

Reviewer #1: Yes

3. Has the statistical analysis been performed appropriately and rigorously? 

Reviewer #1: Yes

4. Have the authors made all data underlying the findings in their manuscript fully available?

Reviewer #1: Yes

5. Is the manuscript presented in an intelligible fashion and written in standard English?

Reviewer #1: Yes

6. Review Comments to the Author

Reviewer #1: The authors have done an excellent job in addressing all of my earlier comments and suggestions. The overall outline now is coherent and focused. Moreover, important issues concerning methods and results have been clarified. I appreciate the authors' conscientiousness and attentiveness to the revisions. Methods and presentation of results still seem very complex to me, but I acknowledge the efforts to improve the structure of the manuscript. In particular the revised discussion section presents now with a concise focus and characterizes the main findings appropriately. In my opinion, the manuscript provides a significant and essential contribution to our understanding of viewing behavior and evaluates data from three established datasets.

7. PLOS authors have the option to publish the peer review history of their article (what does this mean?). If published, this will include your full peer review and any attached files.

Reviewer #1: **Yes: **Jutta Billino

---

## [Editor Report · Acceptance letter]

13 Sep 2023

PONE-D-22-33922R1 

What Can Entropy Metrics Tell Us About the Characteristics of Ocular Fixation Trajectories? 

Dear Dr. Melnyk:

I'm pleased to inform you that your manuscript has been deemed suitable for publication in PLOS ONE. Congratulations! Your manuscript is now with our production department. 

Kind regards, 

on behalf of

Dr. Patricia Wollstadt 

Academic Editor

PLOS ONE